# Prevention of CaCl$_2$-induced aortic inflammation and subsequent aneurysm formation by the CCL3–CCR5 axis

Yuko Ishida[1], Yumi Kuninaka[1], Mizuho Nosaka[1], Akihiko Kimura[1], Akira Taruya[2], Machi Furuta[3], Naofumi Mukaida [4] & Toshikazu Kondo [1✉]

Inflammatory mediators such as cytokines and chemokines are crucially involved in the development of abdominal aortic aneurysm (AAA). Here we report that CaCl$_2$ application into abdominal aorta induces AAA with intra-aortic infiltration of macrophages as well as enhanced expression of chemokine (C-C motif) ligand 3 (CCL3) and MMP-9. Moreover, infiltrating macrophages express C-C chemokine receptor 5 (CCR5, a specific receptor for CCL3) and MMP-9. Both $Ccl3^{-/-}$ mice and $Ccr5^{-/-}$ but not $Ccr1^{-/-}$ mice exhibit exaggerated CaCl$_2$-inducced AAA with augmented macrophage infiltration and MMP-9 expression. Similar observations are also obtained on an angiotensin II-induced AAA model. Immunoneutralization of CCL3 mimics the phenotypes observed in CaCl$_2$-treated $Ccl3^{-/-}$ mice. On the contrary, CCL3 treatment attenuates CaCl$_2$-induced AAA in both wild-type and $Ccl3^{-/-}$ mice. Consistently, we find that the CCL3–CCR5 axis suppresses PMA-induced enhancement of MMP-9 expression in macrophages. Thus, CCL3 can be effective to prevent the development of CaCl$_2$-induced AAA by suppressing MMP-9 expression.

[1] Department of Forensic Medicine, Wakayama Medical University, Wakayama, Japan. [2] Department of Cardiovascular Medicine, Wakayama Medical University, Wakayama, Japan. [3] Department of Clinical Laboratory Medicine, Wakayama Medical University, Wakayama, Japan. [4] Division of Molecular Bioregulation, Cancer Research Institute, Kanazawa University, Kanazawa, Japan. ✉email: kondot@wakayama-med.ac.jp

An abdominal aortic aneurysm (AAA) is characterized by enlarged aorta with the structural destruction composed of the fragmentation of the extracellular matrix. Several conditions including aging, hypertension, smoking, hypercholesterolemia, diabetes, infection, genetic conditions, and trauma are denoted as risk factors for AAA formation[1,2]. Moreover, males are reported to have a much higher risk of developing AAA than females[3]. AAA usually causes no severe symptoms, at most with faint abdominal, back, or leg pains, and as a consequence, it frequently ruptures without preceding symptoms and progresses rapidly, resulting in high acute mortality up to about 80%[2]. Thus, AAA is one of the major causes of sudden unexpected natural death.

Multiple factors such as atherosclerosis, hemodynamics, genetic deficiency, microbiological infection, trauma, environmental conditions, and immunological states are presumed to be involved in the pathogenesis of AAA. Inflammation is temporally and spatially associated with disruption of the orderly lamellar structure of the aortic media and eventually leads to AAA development and progression. In support of this assumption, human AAA lesions exhibit dense infiltration of a myriad of innate and adaptive immune cells, including macrophages, lymphocytes, neutrophils, mast cells, and dendritic cells[4]. Among these infiltrates, macrophages and T lymphocytes are the predominant cell types[5,6]. Consistently, the intense inflammatory response can cause gradual aortic dilatation and subsequent aneurysm formation in the aorta and carotid artery in animal models[7,8]. Moreover, blocking aortic wall inflammation can limit the development and progression of aortic aneurysm[9], and accentuation of inflammation can increase the frequency and size of the aneurysms[10,11].

Inflammation in the aorta can frequently induce the breakdown of normally long-lived matrix macromolecules present in aortic walls, such as collagen and elastin, and eventually dilate the aortic wall, resulting in aneurysm formation. Collagen and elastin are mainly degraded by a family of endopeptidases, matrix metalloproteinases (MMPs). Moreover, several lines of evidence indicate that MMPs with a high capacity to degrade collagen, particularly MMP-9, was expressed by infiltrating macrophages as well as resident vascular smooth muscle cells in aneurysm lesions[12–15]. Based on these observations, the efficacy of MMP inhibitors was examined in preclinical animal models[16–18] and in early clinical trials for patients with aneurysm[19,20]. However, MMP inhibitors could only delay the speed of aneurysm formation without preventing its progression. Thus, it is necessary to establish an effective therapeutic strategy for aneurysm based on the understanding of the molecular and cellular mechanisms underlying aneurysm formation[21].

The difficulties in obtaining human aneurysm tissues at the early phase have hampered the understanding of the molecular and cellular changes at the early step of aneurysm formation. As a consequence, animal models are still indispensable for clarifying the early step of aneurysm formation, aortic dilatation. Currently, three methods have been commonly used to induce aortic dilatation and subsequent aneurysm formation; periaortic application of $CaCl_2$, transient intraluminal elastase perfusion, and angiotensin II (Ang II)-infusion[22]. In this study, we use mainly periaortic application of $CaCl_2$ but also Ang II-infusion to induce aortic dilatation and aneurysm formation, because these models exhibit abundant inflammatory infiltrates and MMP expression, similarly as observed on human AAA[13].

Chemokine system is essential for leukocyte trafficking and eventual inflammatory responses and tumor development[23]. Moreover, several chemokines could have diverse roles in the aneurysm formation in a context-dependent manner[24–26]. Hoh et al.[25] demonstrated that CCL2 could promote elastase-induced

aneurysm formation of the carotid artery by using the CCL3-mediated pathway. MacTaggart and colleagues[24] revealed the involvement of CCR2 but not CCR5 in $CaCl_2$-induced AAA development in SV129 mice. CCR5 and CCR1 are specific receptors for CCL3, a macrophage-tropic inflammatory chemokine[23]. Our immunohistochemical analysis of human AAA samples detects the expression of CCL3 and CCR5 as well as MMP-9 (Supplementary Fig. 1). Moreover, intra-aortic $MMP9$ mRNA expression is significantly enhanced in human AAA samples, compared with human normal aorta ones (Supplementary Fig. 1). However, it still remains elusive on the pathophysiological roles of CCL3 in AAA formation.

$CaCl_2$ treatment enhances the intra-aortic CCL3 expression in mice, preceding AAA formation. In contrary to our expectation that the lack of CCL3 may prevent the development of an abdominal aneurysm, AAA formation is exaggerated in $Ccl3^{-/-}$ mice, compared with WT mice, with enhanced macrophage infiltration and $Mmp9$ expression in the aorta. Moreover, the absence of CCR5 but not CCR1 also exacerbates $CaCl_2$-induced AAA formation like $Ccl3^{-/-}$ mice. Similar observations are obtained on the Ang II-infusion AAA model. Moreover, CCL3 treatment can prevent $CaCl_2$-induced AAA formation in both wild-type and $Ccl3^{-/-}$ mice with dampened MMP-9 expression. Thus, CCL3, hitherto considered as a typical inflammatory chemokine, can exhibit anti-inflammatory activities by acting its specific receptor, CCR5, in these AAA models.

## Results

**Exaggerated AAA formation in the absence of CCL3.** We initially examined immunohistochemically the expression of CCL3 in human aortic aneurysmal tissues. CCL3 was mainly detected in infiltrating leukocytes present in aortic tissues (Fig. 1a). Double-color immunofluorescence analysis demonstrated that CD68$^+$ macrophages were the main cellular source of CCL3 (Fig. 1a). These observations would indicate the potential involvement of CCL3 in aortic aneurysm formation, probably by regulating macrophages, which express abundantly CCL3. We next examined $Ccl3$ mRNA expression in the aorta of WT mice after $CaCl_2$ treatment. $Ccl3$ mRNA could be faintly detected in the untreated aortic tissue. $CaCl_2$ treatment significantly enhanced $Ccl3$ mRNA expression in the aorta at 1 and 2 weeks, decreasing thereafter (Fig. 1b). Likewise, intra-aortic CCL3 protein contents were transiently increased after $CaCl_2$ treatment (Fig. 1c). Double-color immunofluorescence analysis further detected CCL3 protein mainly in F4/80$^+$ macrophages recruited into the adventitia of the aorta (Fig. 1d). Moreover, apoptotic macrophage numbers were increased simultaneously with a decline in CCL3 expression (Supplementary Fig. 2). Thus, macrophages could produce CCL3 after migrating into $CaCl_2$-induced aneurysm lesions but might lose a capacity to express CCL3 when they became apoptotic. To determine the roles of CCL3 in aneurysm formation, we treated simultaneously WT and $Ccl3^{-/-}$ mice with $CaCl_2$ application into the aorta. Despite no significant differences in the aortic diameter between untreated WT and $Ccl3^{-/-}$ mice, $Ccl3^{-/-}$ mice displayed increased aortic diameter to a larger extent than WT mice 6 weeks after $CaCl_2$ application (Fig. 1e, f and Supplementary Table 1). Moreover, 7 out of 8 $Ccl3^{-/-}$ mice exhibited a more than 35% increase in the aortic diameter. There was no difference in the histological structure of the aorta between untreated WT and $Ccl3^{-/-}$ mice (Fig. 1g, HE). On the contrary, at 6 weeks after $CaCl_2$ application, $Ccl3^{-/-}$ mice displayed disruption and fragmentation of medial elastic fibers, and lamellar of the aortic wall, to a larger extent, than WT mice (Fig. 1g, h). These observations implied that macrophage-derived CCL3 would be protective for $CaCl_2$-induced AAA formation.

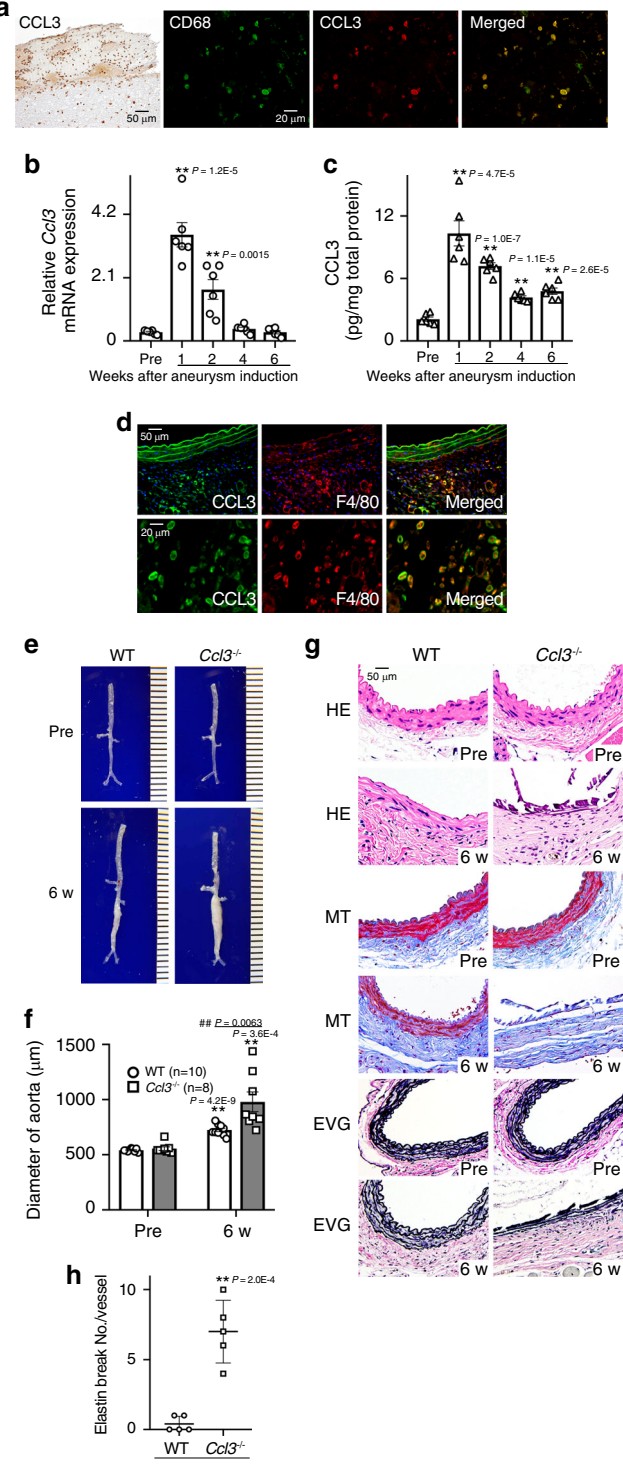

**Fig. 1 Roles of CCL3 in AAA formation. a** CCL3 expression by CD68[+] macrophages in human AAA samples (left panel, scale bar = 50 μm; other panels, scale bar = 20 μm). Representative results from 4 independent experiments are shown here. **b–d** Intra-aortic CCL3 expression in WT mice after CaCl$_2$ treatment. **b** Intra-aortic Ccl3 expression ($n = 6$ in each time point). **$P < 0.01$, vs. pretreatment. **c** Intra-aortic CCL3 protein contents ($n = 6$ in each time point). **$P < 0.01$, vs. pretreatment. **d** Representative images of CCL3 expression by F4/80[+] macrophages in the aortas from WT mice (upper panels, scale bar = 50 μm; lower panels, scale bar = 20 μm). Representative results from four independent experiments are shown here. Blue coloration, nuclear staining by DAPI. **e** Representative macroscopic appearances of aortas in WT and Ccl3$^{-/-}$ mice after CaCl$_2$ treatment (from six independent experiments). **f** Aortic diameters were measured in WT and Ccl3$^{-/-}$ mice ($n = 6$ independent experiments). **$P < 0.01$, vs. pretreatment in each strain; ##$P < 0.01$, vs. WT-posttreatment. **g** Histopathological analysis of the aortas obtained from WT and Ccl3$^{-/-}$ mice after CaCl$_2$ treatment. Representative results from five independent experiments are shown here. Scale bar, 50 μm. HE hematoxylin-eosin staining, MT Masson trichrome staining, EVG Elastica van Gieson staining. **h** Quantification of the number of elastin breaks per vessel. **$P < 0.01$, vs. WT mice. One-way ANOVA followed by Dunnett's post-hoc test was used in (**b**) and (**c**). Unpaired two-sided Student's $t$ test was used in (**f**) and (**h**). Data are presented as mean values ± SEM.

immune cell population between WT and Ccl3$^{-/-}$ mice (Supplementary Fig. 3a, c–g). Treatment with CaCl$_2$ enhanced the recruitment of F4/80[+] macrophages into the adventitia of the aorta in Ccl3$^{-/-}$ mice, more markedly than in WT ones (Fig. 2a, b), but without any significant differences in CD3[+] T cell, CD4[+] T cell, CD8[+] T cell, and B220[+] B cell numbers in the adventitia of the aorta between WT and Ccl3$^{-/-}$ mice (Fig. 2c, d, Supplementary Fig. 4). We next examined intra-aortic gene expression of MMPs and TIMPs, which are presumed to be involved in aortic aneurysm formation[12,13,15,29,30] and observed that Ccl3$^{-/-}$ mice exhibited significantly enhanced intra-aortic Mmp9 mRNA expression compared with WT mice (Fig. 2e). However, there were no significant differences in intra-aortic gene expression of other MMPs and TIMPs between both strains (Supplementary Fig. 5). Consistently, CaCl$_2$ application increased intra-aortic MMP-9 activities to a larger extent in CCL3$^{-/-}$ than WT mice (Fig. 2f). Furthermore, F4/80[+] macrophages infiltrating into aortas expressed MMP-9 (Fig. 2g). In addition, the administration of anti-CCL3 antibody significantly aggravated CaCl$_2$-induced aneurysmal formation in WT mice, together with exaggerated pathological changes, macrophage infiltration, and MMP-9 expression and activity (Fig. 3 and Supplementary Table 2). These observations would indicate that the lack of CCL3 could aggravate CaCl$_2$-induced AAA formation by augmenting macrophage recruitment and macrophage-derived MMP-9 expression.

**Enhanced inflammation and MMP-9 expression in Ccl3$^{-/-}$ mice.** Both in humans and rodents, the development of aneurysm is closely associated with the local inflammatory responses including inflammatory cell infiltration into the adventitia and medium[13]. Exaggerated AAA formation in Ccl3$^{-/-}$ mice prompted us to investigate the inflammatory responses in AAA lesions because CCL3 is a typical inflammatory chemokine acting mainly on inflammatory cells, particularly monocytes/macrophages[27,28]. After the CaCl$_2$ application, the total number of circulating immune cells was increased, to a similar extent, in WT and Ccl3$^{-/-}$ mice (Supplementary Fig. 3a, b), with no significant difference in each

**Involvement of CCL3 produced by bone marrow cells in AAA.** In order to exclude the contribution of non-hematopoietic cell-derived CCL3 to CaCl$_2$-induced aneurysm formation, we next applied CaCl$_2$ in bone marrow (BM) chimeric mice generated between WT and Ccl3$^{-/-}$ mice. The mice transplanted with Ccl3$^{-/-}$ mouse-derived BM cells exhibited exaggerated CaCl$_2$-induced aneurysm formation together with enhanced macrophage infiltration and Mmp9 expression compared with those transplanted with WT mouse-derived BM cells in both WT and Ccl3$^{-/-}$ mice (Fig. 4 and Supplementary Table 3). Thus, BM cell-derived CCL3, probably macrophage-derived one, could be protective for CaCl$_2$-induced AAA formation by suppressing macrophage-derived MMP-9 expression.

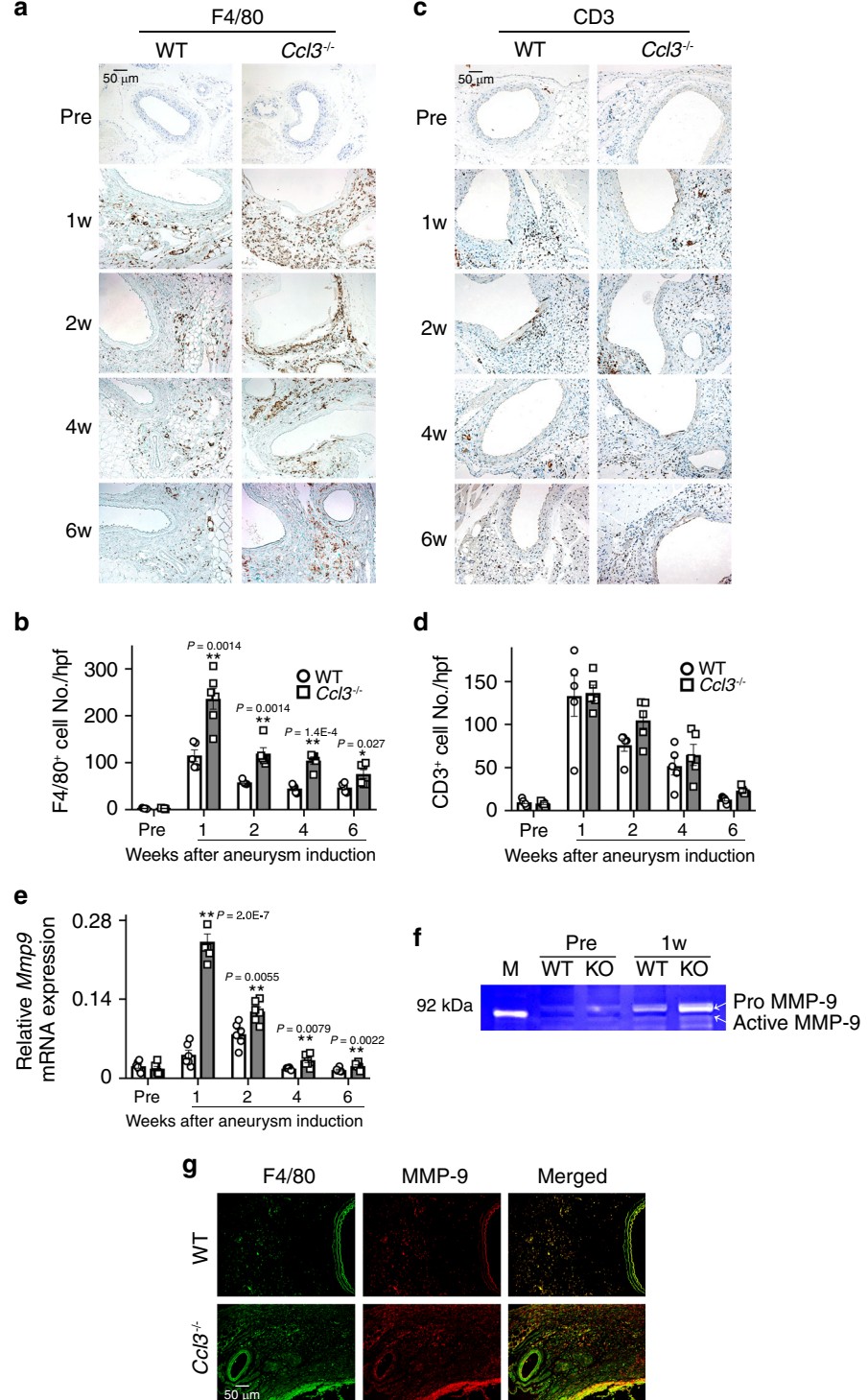

**Fig. 2 Leukocyte recruitment and MMP-9 expression in the aortas after CaCl₂ treatment. a**, **c** The accumulation of macrophages (**a**) and CD3⁺ T cells (**c**) into aortic tissues in WT and *Ccl3⁻/⁻* mice. Representative results from six independent experiments. Scale bar, 50 μm. **b**, **d** The numbers of macrophages (**b**) and CD3⁺ T cells (**d**) were determined. (F4/80: $n = 6$ in 1 week and 6 week of *Ccl3⁻/⁻*, $n = 5$ each in other all groups; CD3: $n = 5$ each in all-time points of both mouse strain). **$P < 0.01$, *$P < 0.05$, vs. WT mice. **e** Intra-aortic gene expression ($n = 5$ each in all-time points of both strains). *$P < 0.05$; **$P < 0.01$, vs. WT mice. **f** Intra-aortic MMP-9 activities at 1 week after CaCl₂ treatment. Representative gel image from four independent experiments. M molecular weight marker. **g** MMP-9 expression by F4/80⁺ macrophages in aortic tissues of WT and *Ccl3⁻/⁻* mice 2 weeks after CaCl₂ treatment. Representative results from four independent experiments. Scale bar, 50 μm. Two-way ANOVA followed by Dunnett's post hoc test was used in (**b**), (**d**), and (**e**). Data are presented as mean values ± SEM.

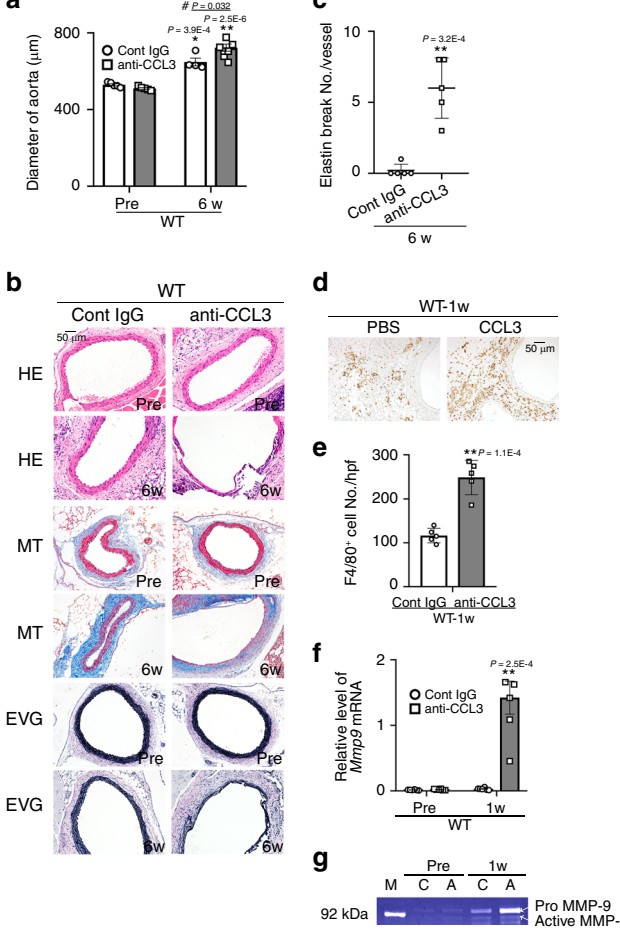

**Fig. 3 Effects of the administration of anti-CCL3 Abs on CaCl2-induced AAA. a** Aortic diameters in mice after CaCl2 treatment. (Pre: $n = 5$ each in control IgG and anti-CCL3 treatment; 6 week: $n = 4$ in control IgG treatment, $n = 8$ in anti-CCL3 treatment). **$P < 0.01$, *$P < 0.05$, vs. pretreatment in each strain; #$P < 0.05$, vs. PBS-treated mice. **b** Histopathological analysis of the aortas obtained from control IgG-treated and anti-CCL3-treated WT mice after CaCl2 treatment. Representative results from 4 independent experiments are shown here. Scale bar, 50 μm. **c** Quantification of the number of elastin breaks per vessel ($n = 5$ each in control IgG and anti-CCL3 treatment). **$P < 0.01$, vs. control IgG. **d** Macrophage accumulation into aortic tissues of control IgG-treated and anti-CCL3 treated mice after CaCl2 treatment. Representative results from five independent experiments are shown. Scale bar, 50 μm. Scale bar, 50 μm. **e** Macrophage numbers were measured ($n = 5$ each in control IgG and anti-CCL3 treatment). **$P < 0.01$ vs. control IgG-treated WT mice. **f** Intra-aortic *Mmp9* expression in IgG-treated and anti-CCL3 treated mice after CaCl2 treatment ($n = 6$ each). **$P < 0.01$, vs. control IgG-treated WT mice. **g** Intra-aortic MMP-9 activities in IgG-treated and anti-CCL3 treated mice at 2 weeks after CaCl2 treatment. Representative gel images from four independent experiments are shown. M molecular weight marker, C control IgG, A anti-CCL3. Unpaired two-sided Student's *t* test was used in (**a**), (**c**), (**e**), and (**f**). Data are presented as mean values ± SEM.

## Lack of CCR5 exaggerates AAA formation

CCL3 utilizes two distinct chemokine receptors, CCR1 and CCR5, as its specific receptors[23]. In order to evaluate the contribution of each receptor, the mice deficient in *Ccr1* or *Ccr5* were treated with CaCl2 along with WT mice. No differences in the diameter of the aorta were observed among these mice when untreated (Fig. 5a and Supplementary Table 4). CaCl2 increased significantly the aortic diameters in each strain at 6 weeks after the treatment, and the

increase, pathological changes, macrophage infiltration, and MMP-9 expression were augmented in $Ccr5^{-/-}$ but not $Ccr1^{-/-}$ mice, compared with WT ones (Fig. 5a–f and Supplementary Table 4). Moreover, we examined in vivo effects of a specific CCR5 antagonist, maraviroc, on aneurysm formation in WT and $Ccl3^{-/-}$ mice. Pharmacological blockage of CCR5 with maraviroc exaggerated CaCl2-induced aneurysm formation in WT mice similarly as observed in $Ccr5^{-/-}$ mice (Fig. 5g). However, maraviroc treatment failed to have any effects on CaCl2-induced aneurysm formation in $Ccl3^{-/-}$ mice (Fig. 5g). Moreover, CCR5 was expressed by macrophages in the aorta of CaCl2-treated WT mice (Fig. 5h). Thus, these observations implied that the interaction of CCL3 with CCR5 but not CCR1 can prevent CaCl2-induced AAA formation.

**Protective effects of an MMP-9 inhibitor on AAA.** In line with the previous studies[16–18], we observed that the enhanced expression and activation of MMP-9 were closely associated with augmented aneurysm formation, indicating the crucial involvement of MMP-9 in aneurysm formation. Hence, we examined the in vivo effects of an MMP-9 inhibitor on aneurysm formation in WT, $CCl3^{-/-}$ and $Ccr5^{-/-}$ mice after CaCl2 treatment. The administration of an MMP-9 inhibitor abrogated CaCl2-treatment-mediated enhancement in aortic diameters in $CCl3^{-/-}$ and $Ccr5^{-/-}$ mice as well as WT mice (Fig. 5i). These observations implied that MMP-9 would be a key molecule for the development of CaCl2-induced AAA formation.

**Macrophage polarization in AAA formation.** Several lines of evidence implied that macrophage polarization, particularly a higher M1/M2 ratio, was associated with AAA formation[29,31]. However, CaCl2 application upregulated the intra-aortic gene expression of both *Nos2* (M1 marker) and *Cd206* (M2 marker) in WT and $Ccl3^{-/-}$ mice. The absence of CCL3 further enhanced *Nos2* gene expression, compared with WT mice (Fig. 6a), but reduced *Cd206* mRNA expression (Fig. 6b). Immunohistochemical analyses demonstrated that $Ccl3^{-/-}$ mice exhibited a higher ratio of NOS2+ M1-macrophages to CD206+ M2-macrophages in the aorta than WT ones (Fig. 6c, d). CCR5 was expressed on both NOS2+ M1-macrophages and CD206+ M2-macrophages (Fig. 6c, e) but these observations implied that CCL3 deficiency could favor macrophage polarization into M1-like phenotype, thereby contributing to AAA formation.

**Protective effects of CCL3 on AAA formation.** In order to examine the preventive effects of CCL3, CCL3 was continuously administered to WT, $Ccl3^{-/-}$, or $Ccr5^{-/-}$ mice, beginning immediately after the CaCl2 application. CCL3 treatment abrogated AAA formation in WT and $Ccl3^{-/-}$ but not $Ccr5^{-/-}$ mice (Fig. 7a–c, g–i, m–o, and Supplementary Table 5). Moreover, attenuated aneurysm formation was accompanied by depressed macrophage infiltration and MMP-9 expression (Fig. 7d–f, j–l and p–r). These observations would imply that CCL3 can dampen CaCl2-induced AAA formation by suppressing MMP-9 expression by macrophages.

**Role of CCL3–CCR5 axis on MMP-9 expression in macrophages.** Enhanced gene expression of *Mmp9* in $Ccl3^{-/-}$ mice, incited us to evaluate the effects of CCL3 on *Mmp9* expression on macrophages in vitro. PMA treatment augmented *Mmp9* mRNA expression in murine peritoneal macrophages (Fig. 8a, b), while CCL3 significantly suppressed PMA-induced gene expression of *Mmp9* in a dose-dependent manner (Fig. 8a). Moreover, CCL3 suppressed PMA-induced *Mmp9* gene expression in THP-1 cells, a human macrophage cell line (Supplementary Fig. 6). Furthermore, these effects were not observed on peritoneal macrophages derived

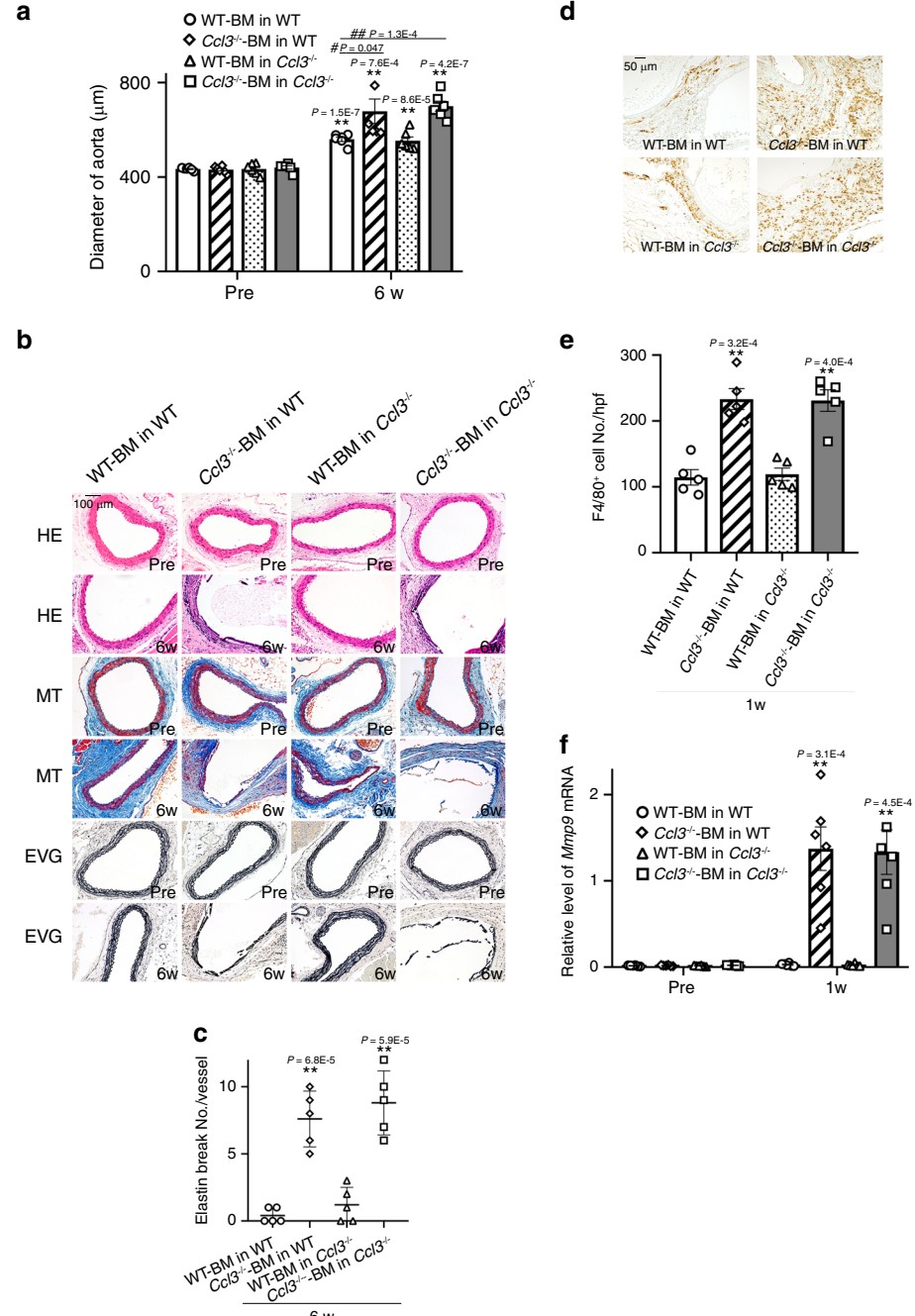

**Fig. 4 Effects of BM transplantation between WT and *Ccl3*$^{-/-}$ mice on CaCl$_2$-induced AAA formation. a** Aortic diameters in BM chimeric mice after CaCl$_2$ treatment ($n = 6$ in each group). **$P < 0.01$, vs. pretreatment in each strain; ##$P < 0.01$, #$P < 0.05$, vs. CaCl$_2$-treated WT mice transplanted WT-BM cells. **b** Histopathological analysis of the aortas obtained from BM chimeric mice after CaCl$_2$ treatment. Representative results from four independent experiments are shown. Scale bar, 100 μm. **c** Quantification of the number of elastin breaks per vessel ($n = 5$ in each mouse strain). **$P < 0.01$, vs. WT-BM in WT. **d** Macrophage accumulation into aortic tissues of BM chimeric mice after CaCl$_2$ treatment. Representative results from five independent experiments are shown. Scale bar, 50 μm. **e** Macrophage numbers were measured ($n = 5$ in each mouse strain). **$P < 0.01$, vs. WT mice transplanted WT-derived BM cells, by unpaired two-sided Student's $t$ test. **f** Intra-aortic *Mmp9* expression in BM chimeric mice after CaCl$_2$ treatment ($n = 6$ in each mouse strain). **$P < 0.01$, vs. pretreatment in each strain. Unpaired two-sided Student's $t$ test was used in (**a**), (**c**), (**e**), and (**f**). Data are presented as mean values ± SEM.

from *Ccr5*$^{-/-}$ but not *Ccr1*$^{-/-}$ mice (Fig. 8b). The potential capacity of p38 MAPK to inhibit LPS-induced *Mmp9* expression[32] prompted us to evaluate MAPK signaling pathways. Indeed, CCL3 enhanced PMA-induced p38 MAPK phosphorylation but decreased ERK phosphorylation in macrophages (Fig. 8c–f). Moreover, a specific p38 MAPK inhibitor, SB203580, but not an ERK inhibitor, PD98059, abrogated CCL3-mediated suppression of PMA-induced *Mmp9*

expression (Fig. 8g). Thus, the CCL3–CCR5 axis could depress *Mmp9* gene expression by activating p38 MAPK in macrophages. When CaCl$_2$-treated WT mice were administered with SB239063, aneurysm formation was significantly exaggerated, compared with vehicle treatment (Fig. 8h). Thus, the activation of p38 MAPK through the CCL3–CCR5 axis can be protective for CaCl$_2$-induced aneurysm formation.

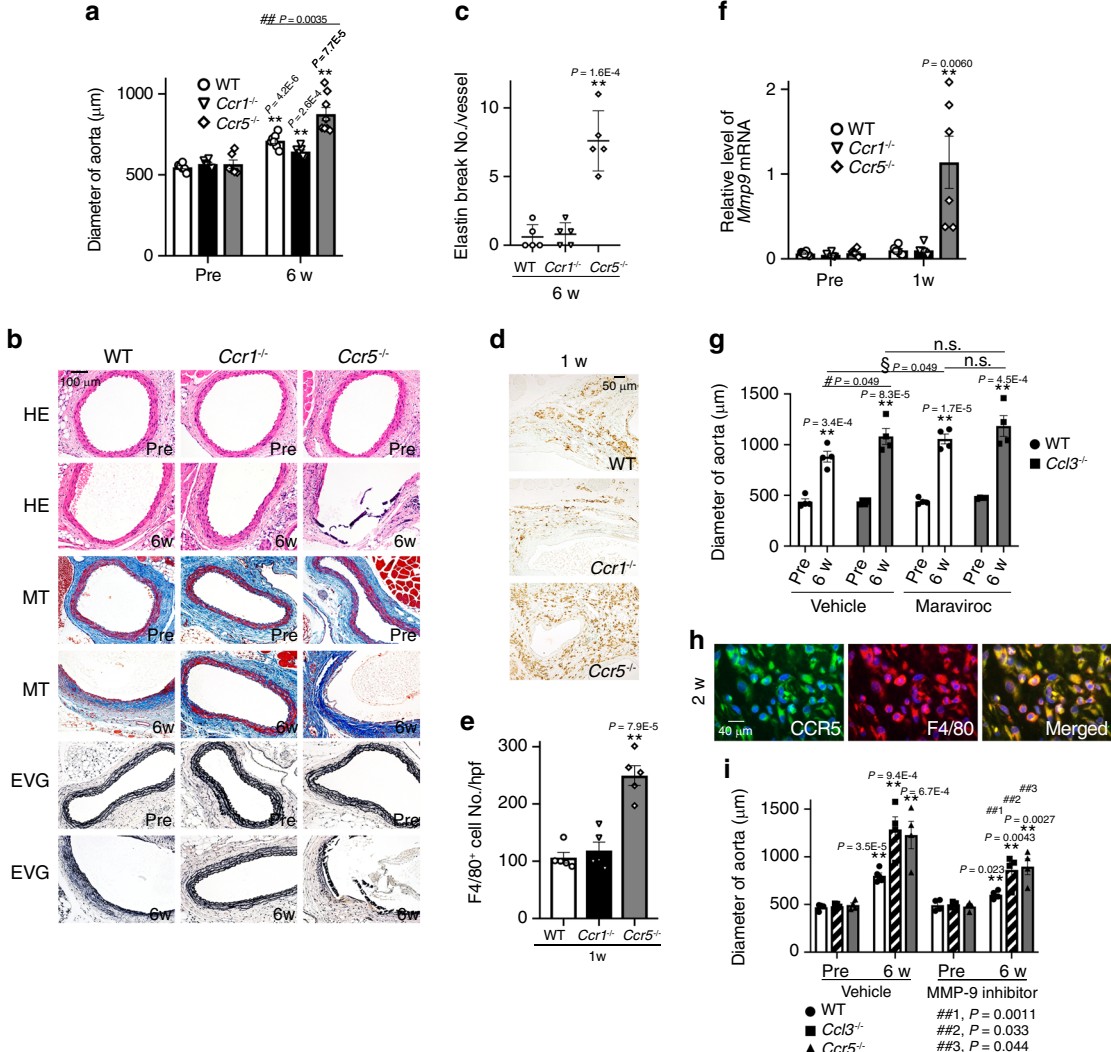

**Fig. 5 The roles of CCL3 receptors in CaCl$_2$-induced AAA formation. a** Aortic diameters in WT, $Ccr1^{-/-}$, and $Ccr5^{-/-}$ mice after CaCl$_2$ treatment. (Pre: $n = 6$ in each strain; 6 week- $n = 7$ each in WT and $Ccr1^{-/-}$, $n = 8$ in $Ccr5^{-/-}$). **$P < 0.01$, vs. pretreatment; ##$P < 0.01$, vs. WT mice. **b** Histopathological analysis of the aortas obtained from WT, $Ccr1^{-/-}$, and $Ccr5^{-/-}$ mice after CaCl$_2$ treatment. Representative results from four independent experiments are shown here. Scale bar = 100 μm. **c** Quantification of the number of elastin breaks per vessel ($n = 5$ in each mouse strain). **$P < 0.01$, vs. WT mice. **d** Macrophage accumulation into aortic tissues of WT, $Ccr1^{-/-}$, and $Ccr5^{-/-}$ mice after CaCl$_2$ treatment. Representative results from five independent experiments are shown. Scale bar, 50 μm. **e** Macrophage numbers were measured ($n = 5$ in each mouse strain). **$P < 0.01$, vs. WT mice. **f** Intra-aortic $Mmp9$ expression in WT, $Ccr1^{-/-}$, and $Ccr5^{-/-}$ mice after CaCl$_2$ treatment ($n = 6$ in each mouse strain). **$P < 0.01$, vs. pretreatment in each strain. **g** Aortic diameters in CaCl$_2$-treated mice after maraviroc application ($n = 4$ each). **$P < 0.01$, vs. pretreatment in each group; #$P < 0.05$, vehicle-treated WT vs. vehicle-treated $Ccl3^{-/-}$WT mice; §$P < 0.05$, maraviroc-treated WT vs. vehicle-treated WT. **h** CCR5-expressing cells in aortic tissues of WT mice after CaCl$_2$ treatment. Representative results from four independent experiments are shown. Scale bar = 40 μm. Blue, nuclear staining by DAPI. **i** Aortic diameters in CaCl$_2$-treated mice after an MMP-9 inhibitor application ($n = 5$ in vehicle-6 week for WT, $n = 4$ each in other all groups). **$P < 0.01$, vs. pretreatment in each group; ##$P < 0.01$, vs. vehicle-treated mice of each strain. Unpaired two-sided Student's $t$ test was used in (**a**), (**c**), (**e**), (**f**), (**g**), and (**i**). Data are presented as mean values ± SEM.

**Roles of CCL3–CCR5 axis in angiotensin II-induced AAA.** In order to validate the roles of the CCL3–CCR5 axis in aneurysm formation in general, we utilized another mouse aortic aneurysm model, where angiotensin II (Ang II) was continuously administered[22,33]. The continuous Ang II infusion elevated systolic blood pressure to similar extents in WT, $Ccl3^{-/-}$, $Ccr1^{-/-}$, and $Ccr5^{-/-}$ mice (Fig. 9a). At 4 weeks after Ang II infusion, we measured the maximal external aortic diameter ex vivo. The aortic diameter was increased to a similar extent in WT and $Ccr1^{-/-}$ mice (Fig. 9b, c), while $Ccl3^{-/-}$ and $Ccr5^{-/-}$ mice exhibited more exaggerated increases in the aortic diameter, compared with WT and $Ccr1^{-/-}$ ones (Fig. 9b, c, Supplementary Fig. 7a). Moreover, the incidence of WT mice and that of $Ccr1^{-/-}$ ones were 8.3% (1 out of 12 in WT

mice) and 10% (1 out of 10 in $Ccr1^{-/-}$ mice), respectively, and there was no significant difference between WT and $Ccr1^{-/-}$ mice (Fig. 9d). On the contrary, the incidences of AAA development in both $Ccl3^{-/-}$ and $Ccr5^{-/-}$ mice were significantly higher, compared with WT ones (64.3%, 9 out of 14 in $Ccl3^{-/-}$ mice; 66.7%, 8 out of 12 in $Ccr5^{-/-}$ mice) (Fig. 9d). However, we never observed the aortic rupture in each mouse strain after Ang II infusion. Histopathological analyses revealed more extended destruction of aortic structure in $Ccl3^{-/-}$ and $Ccr5^{-/-}$ mice, than in WT and $Ccr1^{-/-}$ ones (Fig. 9e). We further examined intra-aortic MMP-9 activities and observed that $Ccl3^{-/-}$ and $Ccr5^{-/-}$ mice, but not $Ccr1^{-/-}$ mice, exhibited significantly enhanced intra-aortic MMP-9 activities, compared with WT mice (Fig. 9f). Furthermore, Ang II application

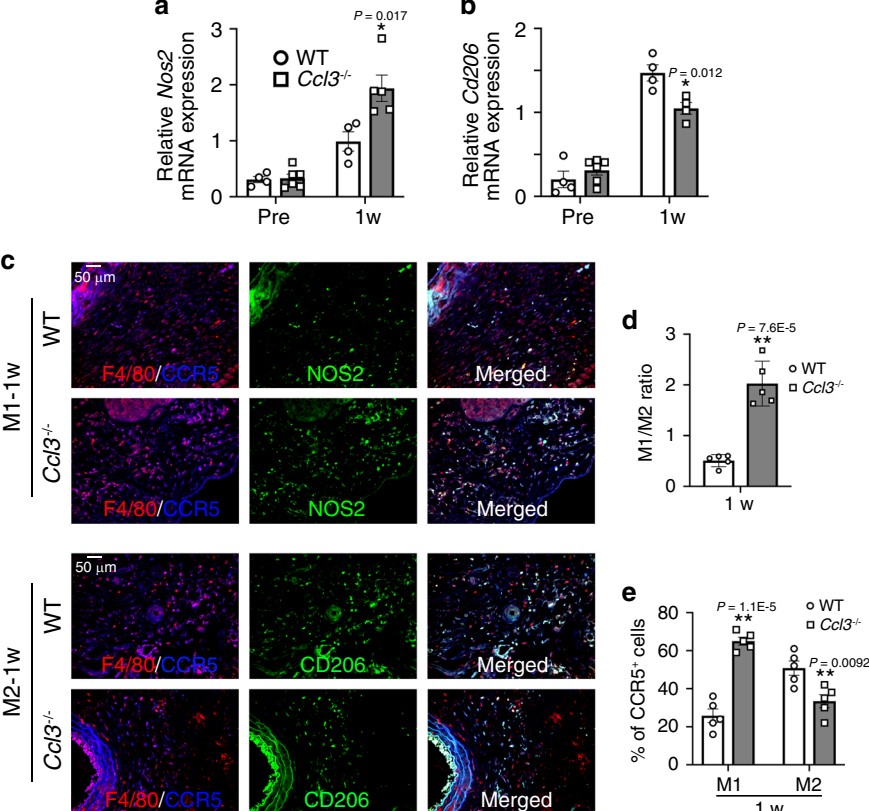

**Fig. 6 Determination of M1- and M2-like phenotypes in macrophages in aortic tissues of WT and *Ccl3*$^{-/-}$ mice after CaCl$_2$ treatment. a, b** Intra-aortic gene expression of **a** *Nos2* (Pre: $n = 4$ in WT, $n = 6$ in *Ccl3*$^{-/-}$; 1 week- $n = 4$ in WT, $n = 5$ in *Ccl3*$^{-/-}$) and **b** *Cd206* (Pre: $n = 4$ in WT, $n = 6$ in *Ccl3*$^{-/-}$; 1 week- $n = 4$ each in WT and *Ccl3*$^{-/-}$). **$P < 0.01$; *$P < 0.05$, vs. WT mice. **c** Triple-color immunofluorescence analysis of F4/80$^+$CCR5$^+$NOS2$^+$ M1 macrophages and F4/80$^+$CCR5$^+$CD206$^+$ M2 macrophages in the AAA tissues of WT and *Ccl3*$^{-/-}$ mice after CaCl$_2$ application. Representative images from five independent experiments are shown. Scale bar, 50 μm. **d** The ratio of M1 to M2 macrophages in WT and *Ccl3*$^{-/-}$ mice ($n = 5$ each in WT and *Ccl3*$^{-/-}$). **$P < 0.01$, vs. WT. **e** The percentage of CCR5$^+$ cells in M1 or M2 macrophages from WT and *Ccl3*$^{-/-}$ mice after CaCl$_2$ application ($n = 5$ each). **$P < 0.01$, vs. WT. Unpaired two-sided Student's $t$ test was used in (**a**), (**b**), (**d**), and (**e**). Data are presented as mean values ± SEM.

increased the numbers of MMP-9$^+$ macrophages which infiltrated into aortas, in *Ccl3*$^{-/-}$ and *Ccr5*$^{-/-}$ mice to a larger extent, compared with WT ones (Supplementary Fig. 7b, d). Altogether, the CCL3–CCR5 axis can have a protective role also in another AAA model, the Ang II-infused aneurysm formation model.

## Discussion

Accumulating evidence implied the pathophysiological roles of the chemokine system in the development of AAA[24–26], but there are still discrepancies in the roles of each chemokine in the aneurysm formation. MacTaggart and colleagues reported few effects of CCR5 deficiency on CaCl$_2$-induced AAA development in SV129 mouse strain[24], whereas Hoh et al.[25] demonstrated the promotion of elastase-induced carotid aneurysm development by CCL3. Our present study revealed that the CCL3–CCR5 axis played protective roles in the development of CaCl$_2$-induced aneurysm. These discrepancies may arise from the differences in the used aneurysm models and mouse strains.

AAAs are characterized by structural alterations of the aortic wall arising from degradation of the macromolecules, collagen, and elastin. Studies of human AAA lesions demonstrated excessive MMP and TIMP expression[1,29,30,34–36], indicating the crucial roles of MMPs in AAA development. Supporting this notion, inhibition of MMPs[16,18–20] could delay the progress of experimental AAA. In the present study, the absence of CCL3 exhibited enhanced expression and activity of MMP-9. However, the gene expression of other MMPs and TIMPs was upregulated to a

similar extent in WT and *Ccl3*$^{-/-}$ mice. These observations implied that CCL3 deficiency might enhance selectively MMP-9 expression.

Simultaneously with MMP overexpression, abundant inflammatory cell infiltration was frequently observed in human AAA lesion[5,6]. These observations further suggest the association of matrix destruction with inflammatory cell infiltration but the results on animal studies are inconsistent in terms of a causal role of local inflammation in AAA pathogenesis[7,13]. Here, we revealed that CaCl$_2$ application induced abundant infiltration of macrophages, which expressed MMP-9, an MMP with a high capacity to degrade collagen. These observations would imply that macrophages could contribute to AAA development by producing MMP-9[37].

It is still controversial the cellular source of MMP-9 in the course of AAA development. Macrophages could enhance MMP-9 production by human vascular smooth muscle cells[37,38], by providing various pro-inflammatory cytokines, which were detected within AAA tissue[5,39]. This assumption may be supported by the observation that TNF-α and IL-1β expression was detected in the aorta of CaCl$_2$-treated mice (Supplementary Fig. 8a–d). The analysis using BM chimeric mice revealed that aneurysm formation and MMP-9 expression were enhanced by CCL3 deficiency restricted to BM-derived cells. Moreover, among various inflammatory cells, only intra-aortic macrophage numbers changed in parallel with MMP-9 expression in the aorta in WT and *Ccl3*$^{-/-}$ mice treated with CaCl$_2$. Furthermore, we revealed that infiltrating macrophages

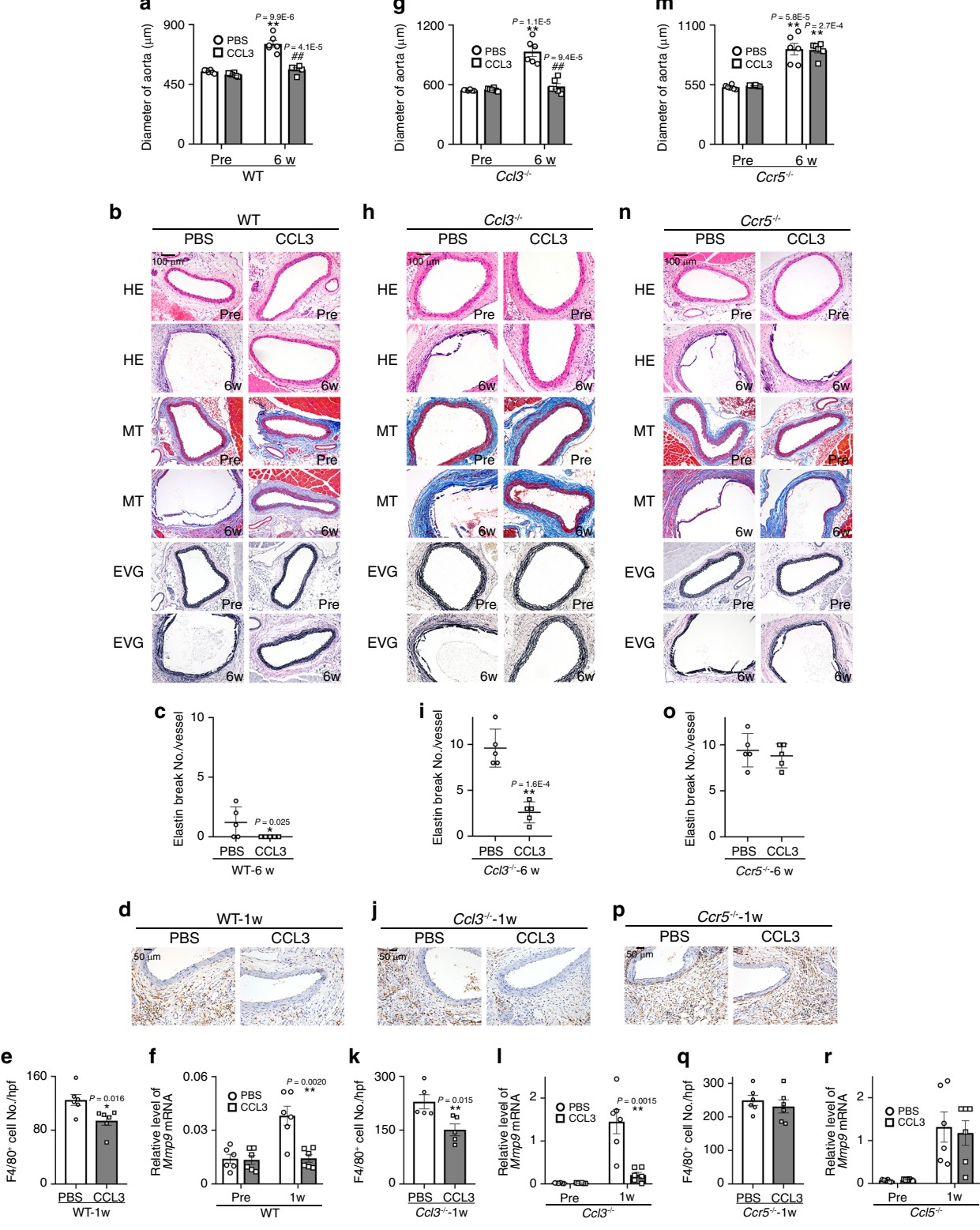

**Fig. 7 Effects of CCL3 on CaCl$_2$-induced AAA formation in WT, *Ccl3*$^{−/−}$, or *Ccr5*$^{−/−}$ mice. a, g, m** Aortic diameters before and after CaCl$_2$ treatment ($n = 6$ each in all groups). **$P < 0.01$, *$P < 0.05$, vs. pretreatment in each strain; ##$P < 0.01$, CCL3 vs. PBS treatment. **b, h, n** Histopathological analysis of the aortas obtained from WT, *Ccl3*$^{−/−}$, and *Ccr5*$^{−/−}$ mice after CaCl$_2$ treatment. Representative results from five independent experiments are shown here. Scale bars, 100 μm. **c, i, o** Quantification of the number of elastin breaks per vessel ($n = 5$ each in all groups). *$P < 0.05$, **$P < 0.01$, vs. PBS. **d, j, p** The effects of CCL3 on macrophage accumulation into aortic tissues after CaCl$_2$ treatment. Representative results from six independent experiments are shown. Scale bars, 50 μm. **e, k, q** Macrophage numbers were measured. (**e, q**: $n = 6$ samples in all groups; **k**: $n = 5$ samples in all groups). **$P < 0.01$, *$P < 0.05$, vs. PBS-treated mice. **f, l, r** Intra-aortic *Mmp9* expression in CaCl2-treated WT, *Ccl3*$^{−/−}$ or *Ccr5*$^{−/−}$ mice after CCL3 treatment ($n = 6$ samples in all groups). **$P < 0.01$, vs. PBS-treated mice. Unpaired two-sided Student's $t$ test was used in (**a**), (**c**–**f**), (**g**), (**i**–**l**), (**m**), and (**o**–**r**). Data are presented as mean values ± SEM.

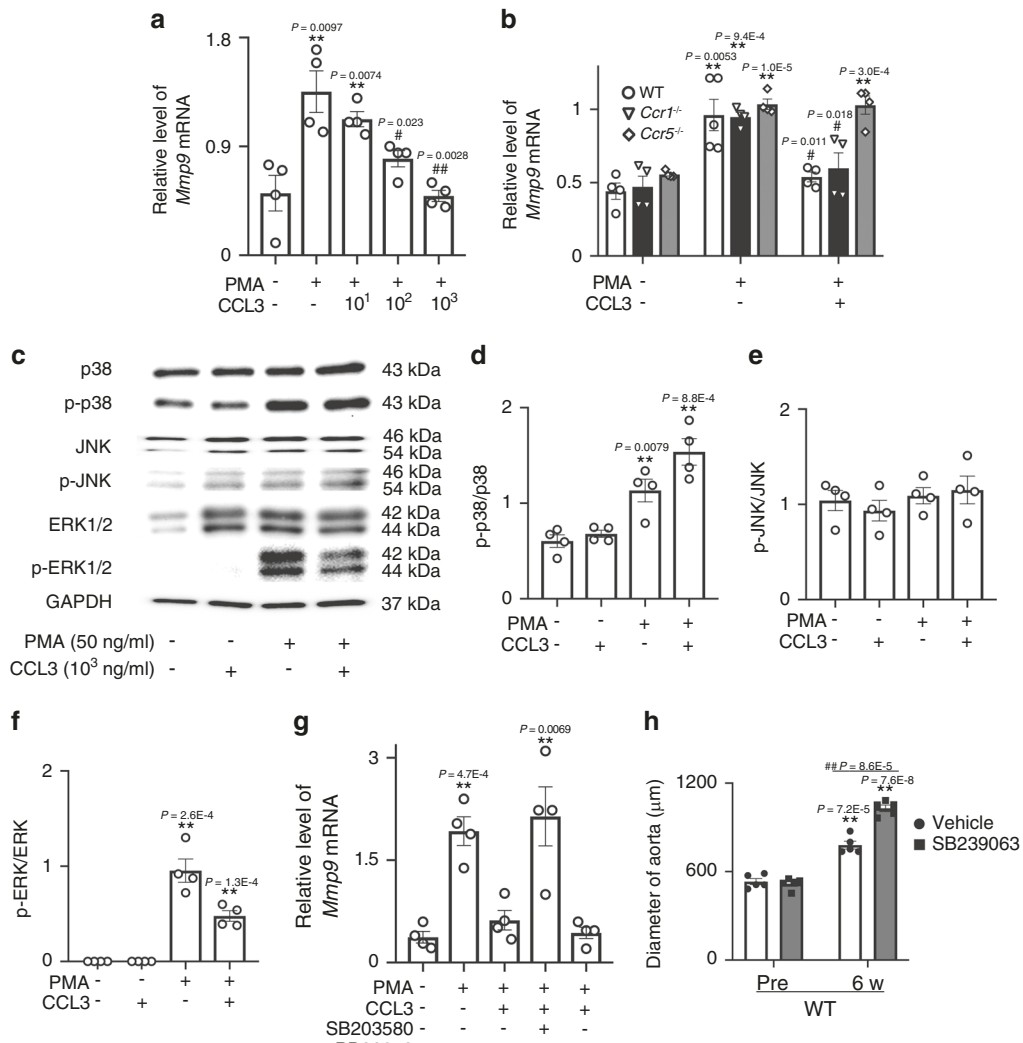

**Fig. 8 Regulation of MMP-9 expression in peritoneal macrophages by CCL3 and its receptors. a** The effects of CCL3 on *Mmp9* expression in WT macrophages stimulated by PMA for 24 h ($n = 4$ independent experiments). **$P < 0.01$, *$P < 0.05$, vs. no stimulation; ##$P < 0.01$, #$P < 0.05$, vs. PMA only. **b** The effects of CCL3 on PMA-induced *Mmp9* expression in peritoneal macrophages obtained from WT, *Ccr1*$^{-/-}$, and *Ccr5*$^{-/-}$ mice ($n = 4$ independent experiments). **$P < 0.01$, vs. no stimulation; ##$P < 0.01$, #$P < 0.05$, vs. PMA only. **c** Western blotting analyses on the expression of ERK, p-ERK, p38, p-p38, JNK, p-JNK, and GAPDH in WT macrophages stimulated by PMA (50 ng/ml) and/or CCL3 ($10^3$ ng/ml) for 24 h. Representative images from four independent experiments are shown. **d–f** Semi-quantitation was performed on the band intensities: **d** p-p38/p38 ratios, **e** p-JNK/JNK ratios, and **f** p-ERK/ERK ratios ($n = 4$ independent experiments). **$P < 0.01$, *$P < 0.05$, vs. no stimulation. **g** The effects of a p38 inhibitor (SB203580) or an ERK inhibitor (PD98059) on *Mmp9* expression in WT macrophages stimulated by PMA and/or CCL3 for 24 h ($n = 4$ independent experiments). **$P < 0.01$, vs. no stimulation. **h** The effects of a p38 inhibitor on CaCl$_2$-induced AAA formation. Aortic diameters were measured in vehicle-treated and SB239063-treated mice before and 6 week after CaCl$_2$ application. (Pre: $n = 5$ each; 6 week- $n = 5$ each in-vehicle and). **$P < 0.01$, vs. pretreatment in each group; ##$P < 0.01$, SB239063 vs. vehicle. One-way ANOVA followed by Dunnett's post hoc test was used in (**a**), (**b**), and (**d–g**). Unpaired two-sided Student's *t* test was used in (**h**). Data are presented as mean values ± SEM.

were the main cellular source of MMPs in the aortic wall of CaCl$_2$-treated mice. These observations would imply an important contribution of macrophage-derived MMP-9 to CaCl$_2$-induced AAA formation. This assumption may be supported by the previous study that MMP-9 was predominantly derived from macrophages present in AAA[40].

It remains elusive how MMP-9 can contribute to AAA formation. Pyo et al.[12] revealed that MMP-9, particularly macrophage-derived one, destructed local matrix proteins and eventually induced aneurysm development in the elastase AAA infusion model. Moreover, *Mmp9*$^{-/-}$ mice were resistant to CaCl$_2$-induced aneurysm development, together with preserved lamellar morphology of aortic walls, compared with WT mice. Longo et al.[13] further proved that macrophage-derived MMP-9 cooperatively induced the development

of experimental AAA in mice. We observed that CaCl$_2$ application enhanced intra-aortic MMP-9 expression. Thus, enhanced activities of MMP-9 can promote AAA formation upon the CaCl$_2$ application. Supporting this notion, an MMP-9 inhibitor significantly suppressed CaCl$_2$-induced AAA formation in WT, *Ccl3*$^{-/-}$, and *Ccr5*$^{-/-}$ mice.

We demonstrated that the CaCl$_2$ application augmented the expression of CCL3, hitherto considered as an inflammatory chemokine with a potent macrophage chemotactic activity, in the adventitia of aorta. However, to our surprise, *Ccl3*$^{-/-}$ mice exhibited exaggerated CaCl$_2$-induced AAA formation with augmented macrophage infiltration and MMP-9 expression in the aorta. Moreover, anti-CCL3 Abs induced similar phenotypes in WT mice, whereas CCL3 administration attenuated CaCl$_2$-induced AAA formation. Furthermore, CCL3 reduced in vitro

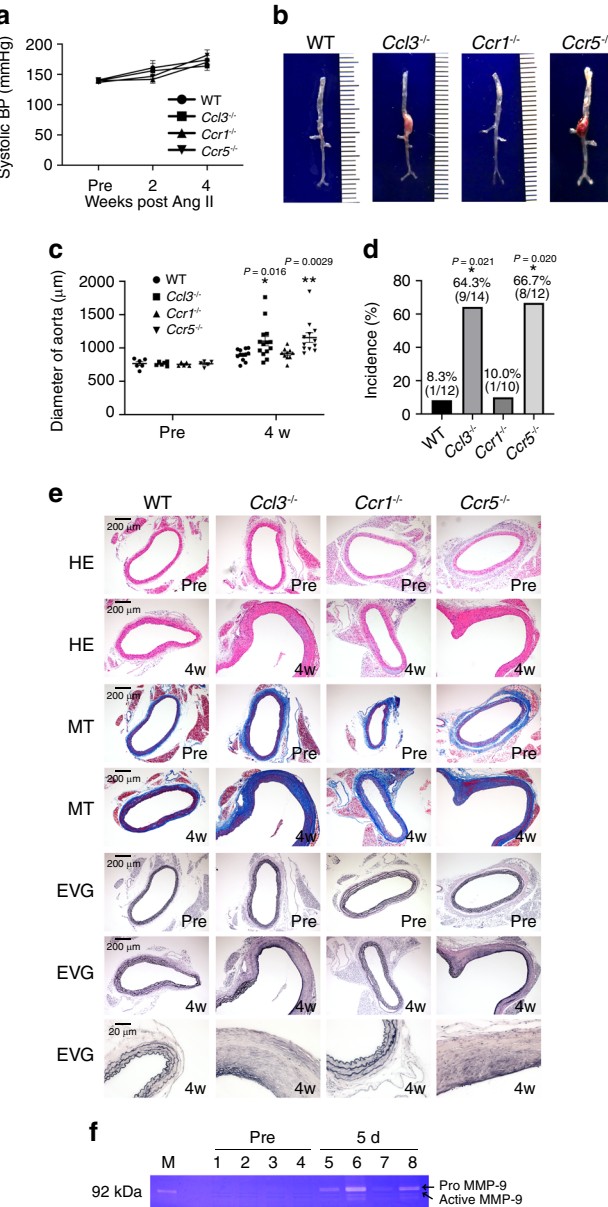

**Fig. 9 The roles of the CCL3–CCR5 axis on Ang II-induced AAA formation. a** Systolic blood pressure at the indicate time points after Ang II infusion (Pre: $n = 7$ in each mouse strain; 2 week: $n = 9$ in WT mice, $n = 4$ each in $Ccl3^{-/-}$ and $Ccr1^{-/-}$, $n = 5$ in $Ccr5^{-/-}$ mice; 4 week: $n = 4$ each in WT, $Ccr1^{-/-}$, and $Ccr5^{-/-}$ mice, $n = 5$ in $Ccl3^{-/-}$ mice). **b** Representative macroscopic appearance of the aorta in each mouse strain after Ang II infusion. **c** The diameters of the aorta in mice at 4 weeks after Ang II infusion (Pre: $n = 6$ each in WT and $Ccl3^{-/-}$ mice, $n = 5$ each in $Ccr1^{-/-}$ and $Ccr5^{-/-}$ mice; 4 week: $n = 12$ each in WT and $Ccr5^{-/-}$ mice, $n = 14$ in $Ccl3^{-/-}$ mice, $n = 10$ in $Ccr1^{-/-}$ mice). **$P < 0.01$, *$P < 0.05$, vs. posttreatment in WT mice. **d** Incidence of AAA in WT, $Ccl3^{-/-}$, $Ccr1^{-/-}$, and $Ccr5^{-/-}$ mice after Ang II infusion. *$P < 0.05$, vs. WT. **e** Histopathological analysis of the aortas obtained from WT, $Ccl3^{-/-}$, $Ccr1^{-/-}$, and $Ccr5^{-/-}$ mice after Ang II application· **f** Intra-aortic MMP-9 activities were measured by gelatin zymography on day 5 after Ang II infusion. M molecular weight marker. 1 and 5, WT; 2 and 6, $Ccl3^{-/-}$; 3 and 7, $Ccr1^{-/-}$; 4 and 8, $Ccr5^{-/-}$. Independent experiments were repeated four times. Unpaired two-sided Student's $t$ test was used in (**a**) Steel–Dwass's multiple comparison test was used in (**d**). Data are presented as mean values ± SEM.

PMA-induced MMP-9 expression in both human and mouse macrophages. These observations would indicate that CCL3 can dampen inflammatory responses in CaCl₂-induced AAA formation, by suppressing MMP-9 expression in macrophages.

CCL3-induced depressed MMP-9 expression was observed only in the presence of CCR5 but not CCR1, among specific receptors for CCL3. Thus, CCL3–CCR5 interaction can have a negative impact on MMP-9 expression in mouse macrophages. Although it remains an open question on the CCL3–CCR5 axis-mediated signal pathways, we revealed that CCL3 could augment PMA-induced p38 MAPK but rather reduced PMA-induced ERK phosphorylation in macrophages. Given the fact that p38 MAPK activation could depress lipopolysaccharide-induced MMP-9 expression in astrocytes[32], we further examined the effect of a p38 MAPK or an ERK inhibitor on CCL3-mediated attenuation of PMA-induced *Mmp9* expression. Indeed, inhibition of p38 MAPK but not ERK pathway reversed CCL3-mediated inhibition of PMA-induced *Mmp9* expression in macrophages. Since p38MAPK can counteract ERK1/2 activation[32], CCL3 can dampen MMP-9 expression by activating p38 MAPK and eventually depressing the ERK pathway. This notion can be reinforced by the observation that the administration of a p38 inhibitor alleviated CaCl₂-induced AAA.

The most puzzling observation in the present study is that intra-aortic macrophage infiltration was augmented in the absence or the inhibition of the CCL3–CCR5 axis, which can exhibit a potent in vitro monocyte/macrophage chemotactic activity[41–43]. Our previous study suggested that macrophages were recruited independently of CCR5-mediated signals[44]. However, the evidence is accumulating to indicate that leukocyte extravasation requires the degradation of matrix proteins present in the basement membrane in the vasculature by various proteinases, particularly MMPs. Indeed, the recruitment of leukocytes such as neutrophils and macrophages was impaired in *Mmp9*-deficient mice[45,46]. Thus, the absence of the CCL3–CCR5 pathway may enhance the expression of MMP-9, which can facilitate macrophage infiltration. Moreover, $Ccl3^{-/-}$ mice exhibited enhanced intra-aortic gene expression of *Ccl2*, a potent macrophage-tropic chemokine, compared with WT mice (Supplementary Fig. 8e), but CCL3 failed to induce in vitro *Ccl2* gene expression in macrophages (Supplementary Fig. 8f). Thus, the enhanced CCL2 expression in $Ccl3^{-/-}$ mice can arise from an increase in recruited macrophages. Collectively, augmented MMP-9 activity in the absence of CCL3 increased macrophage recruitment, and CCL2 produced by the recruited macrophages further increased macrophage infiltration.

Macrophage polarization, particularly, skewed polarization into the M1-like phenotype was associated with AAA formation[29,31]. $Ccl3^{-/-}$ mice exhibited enhanced expression of M1 markers such as NOS2, TNF-α (Supplementary Fig. 8a, c), IL-1β (Supplementary Fig. 8b, d), and CCL2 (Supplementary Fig. 8e), and the M1/M2 ratio was significantly higher in $Ccl3^{-/-}$ mice than in WT ones. Although CCR5 was expressed on both M1 and M2 macrophages, these observations implied the CCL3 deficiency skewed macrophage polarization into the M1-like phenotype, thereby contributing to AAA formation.

Several lines of evidence implicate a crucial involvement of several inflammatory mediators including TNF-α[47] and IL-1β[48] in the development of experimental AAA. Moreover, the expression of these mediators was augmented in human AAA walls[49]. Consistently, intra-aortic TNF-α and IL-1β expression was enhanced after CaCl₂ application and was detected mainly in infiltrating macrophages (Supplementary Fig. 8a–d). Moreover, the increments were further augmented in the absence of CCL3.

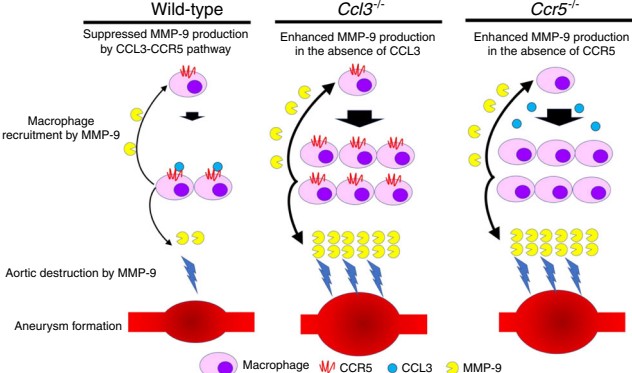

**Fig. 10 Schema of the protective roles of the CCL3–CCR5 pathway in aneurysm formation.** The illustration was created by T.K.

Since the ERK pathway has an indispensable role in macrophage activation, particularly their expression of pro-inflammatory cytokines including TNF-α[50], CCL3-mediated inhibition of the ERK pathway may account for exaggerated intra-aortic TNF-α expression in the absence of CCL3.

Our experimental study demonstrates that the CCL3–CCR5 pathway plays protective roles in AAA formation (Fig. 10). Moreover, a substantial proportion of Caucasians have a 32-base pair deletion allele, called Δ32, in human CCR5 gene[51], and the deletion results in a frame-shift, with a premature stop codon. As a consequence, homozygous persons bearing CCR5 Δ32 mutation do not express functional CCR5 protein similarly as Ccr5$^{-/-}$ mice do not. The association of CCR5 Δ32 mutation with some diseases was reported previously[52–55]. Moreover, CCR5 Δ32 mutation increased the incidence of AAA[56,57], which was consistent with our observations in CaCl2- or Ang II-treated Ccr5$^{-/-}$ mice. Thus, it is reasonable to examine the therapeutic activity of CCL3 for an aneurysm. Furthermore, given that CCL3 or its derivatives have been tried in humans to induce hematopoietic cell mobilization to peripheral blood, without severe adverse effects[58,59], it may be a therapeutic strategy for an aneurysm.

## Methods

**Animals**. Specific pathogen-free 8-week-old male C57BL/6 mice were obtained from Japan SLC and designated as WT mice in this study. Homozygous Ccl3$^{-/-}$ mice were obtained from Jackson Laboratories (Bar Harbor, ME). Ccr1$^{-/-}$ and Ccr5$^{-/-}$ mice a generous gift from Drs. P.M. Murphy and J.L. Gao (National Institute of Allergy and Infectious Diseases, NIH, Bethesda, MD)[60] and K. Matsushima (Division of Molecular Regulation of Inflammatory and Immune Diseases, Research Institute of Biomedical Sciences, Tokyo University of Science, Japan)[61]. All these mice were backcrossed to C57BL/6 mice for ten generations. All mice were kept under the specific pathogen-free conditions at the Animal Research Center of Wakayama Medical University and were housed in plastic cages with wood chips. Age- and sex-matched mice were used for the experiments. All experimental protocols were approved by the Animal Research and Ethics Committee of Wakayama Medical University (approval No. 885). All animal experiments in this study were performed in compliance with the Guidelines for the Care and Use of Laboratory Animals on Wakayama Medical University and with the relevant guidelines and regulations.

**CaCl2-induced AAA model**. AAA was induced by the periaortic application of 0.25 M CaCl2 (Sigma-Aldrich, St. Louis, MO)[47]. Briefly, after deep anesthesia with an intraperitoneal injection of pentobarbital (50 mg/kg body weight), a 2-cm incision was made along the abdominal midline. The abdominal aorta between the renal arteries and bifurcation of the iliac arteries was exposed from the surrounding retroperitoneal structures, and 0.25 M CaCl2 was applied to the external surface of the aorta. After 15 min, the aorta was rinsed with 0.9% sterile saline and the incision was closed, and mice were returned to their cages after recovery. Aortas were obtained to measure their diameters before and 6 weeks after CaCl2 application and were processed to further analyses. The development of AAA was defined as a more than 35% increase relative to the mean aortic diameter of

pretreatment mice in each group. In some experiments, WT mice were i.p. given goat anti-mouse CCL3 pAbs (100 μg/mouse, AB-450-NA, R&D) or normal goat IgG control (50 μg/mouse, AB-108-C, R&D) every week beginning at the day of aneurysm induction for 6 weeks until sacrifice. In another series of experiments, WT mice were i.p. given recombinant CCL3 (2 μg/200 μl/mouse, 450-MA-050, R&D) or PBS in an equal volume twice a week for 6 weeks beginning at the day of CaCl2 application. At the indicated time intervals after the CaCl2 application, mice were sacrificed to obtain aortas for subsequent analyses.

**Ang II infusion-induced AAA model**. Mice were deeply anesthetized and mini osmotic pumps (model 2004; Alzet, Cupertino, CA) were subcutaneously implanted in the neck region of anesthetized mice for 28 days in order to inject constantly Ang II (1 μg/kg/min, Sigma Aldrich)[33]. We employed the following condition as the criteria for Ang II-induced AAA formation: more than 30% increase in the external diameters of the suprarenal aorta, compared with the average of control mice[33]. After we observed the suprarenal abdominal aorta by ultrasound imaging, the aorta samples were obtained for the subsequent measurement of the external diameters of the suprarenal abdominal aortic diameters by an independent researcher ex vivo under a microscope. Since the average of the external diameters of the suprarenal aortic width was 765 μm in control mice, we judged that AAA developed when the external diameters were more than 995 μm.

**Blood pressure measurement**. Blood pressure was measured noninvasively on conscious mice using a CODA volume pressure recording tail-cuff system (Kent Scientific Corporation, Torrington, CT). The systolic blood pressure was measured at least five times, before and 4 weeks after Ang II pump implantation. The mean systolic blood pressure for each group was determined by averaging the systolic blood pressure of each mouse included in that group. The data from the 1-day measurement of each time point were used.

**Transabdominal ultrasound imaging**. For ultrasonic imaging, after removed abdominal hairs using hair removal cream, the animals were placed on a heated (41 °C) imaging stage in the supine position while under anesthesia with ketamine–xylazine. Warmed ultrasound gel was applied to the abdominal surface and ultrasound prove applied to the gelled surface to collect images by the imaging system (APLIO 500 TUS-A500, Toshiba Medical System Corporation, Japan).

**Generation of BM chimeric mice**. The following BM chimeric mice were prepared[62]: male Ccl3$^{-/-}$ BM→female WT mice, male WT BM→female Ccl3$^{-/-}$ mice, male WT BM→female WT mice, and male Ccl3$^{-/-}$ BM→female Ccl3$^{-/-}$ mice. BM cells were collected from the femurs of donor mice by aspiration and flushing. Recipient mice were irradiated to 10 Gy using an RX-650 irradiator (Faxitron X-ray Inc., Wheeling, IL). Then, the animals received intravenously 5 × 10$^6$ donor-derived BM cells in a volume of 200 μl sterile PBS (−) under anesthesia. Thereafter, the mice were housed in sterilized micro isolator cages and were fed normal chow and autoclaved hyperchlorinated water for 60 days. To verify the successful engraftment and reconstitution of the BM in the transplanted mice, genomic DNA was extracted from peripheral blood and tail tissues of each chimeric mouse at 30 days after BM transplantation with a NucleoSpin tissue kit (Macherey-Nagel, Duren, Germany). Then, we performed polymerase chain reaction (PCR) to detect the Sry gene contained in the Y chromosome (forward primer, 5′-TTGCCTCAACAAAA-3′; reverse primer, 5′-AAACTGCTGCTTCTG CTGGT-3′). The amplified PCR products were fractionated on a 2% agarose gel and visualized by ethidium bromide staining. After durable BM engraftment was confirmed, mice were treated with CaCl2 as described above.

**Histopathological and immunohistochemical analyses**. The aortas of humans and mice were fixed with 10% neutral-buffered formalin and were then embedded with paraffin to prepare sections. The deparaffinized sections were subjected to hematoxylin and eosin (HE), Masson trichrome, or Elastica-van Gieson staining for histopathological analysis. Elastin degradation was quantified by counting the number of breaks per vessel and then averaged and graphed. Immunohistochemical analysis was performed[44]. Deparaffinized aorta sections of humans and mice were immersed in 0.3% H2O2 in methanol for 30 min to eliminate endogenous peroxidase activities. The sections were further incubated with PBS containing 1% normal serum corresponding to the secondary Ab and 1% bovine serum albumin to reduce nonspecific reactions. Antigen retrieval was performed by boiling in a commercialized retrieval buffer (901-CB911-071117, Biocare Medical, Pacheco, CA and S1699, DAKO, Glostrup, Denmark). Subsequently, sections were incubated overnight at 4 °C with the following primary Abs: rabbit anti-human CCL3 pAbs (1:50, GTX52609, Gene Tex, Irvine, CA), mouse anti-human CD68 mAb (1:100, clone PG-M1, M0876, DAKO), goat anti-human MMP-9 pAbs (1:50, sc-6840, Santa Cruz Biotechnology, Dallas, TX), goat anti-human CCR5 pAbs (1:300, GTX21673, Gene Tex, Irvine, CA), rat anti-mouse F4/80 mAb (1:50, clone BM8, T-2028, BMA Biomedicals, Switzerland), rat anti-human CD3 mAb which cross-reacts with mouse CD3 (1:70, clone CD3-12, MCA1477, AbD Serotec, Raleigh, NC), rat anti-mouse B220/CD45R mAb (1:100, clone RM0063-9F14, NBP2-12168, Novs Biologicals, Centennial, CO), rabbit anti-mouse CD4 mAb

(1:200, clone EPR19514, ab183685, Abcam, Cambridge, UK), rat anti-mouse CD8 pAbs (1:100, ab203035, Abcam, Cambridge, UK) or rabbit anti-ssDNA pAbs (1:50, IBL, Gunma, Japan). After the incubation with appropriate secondary Abs such as biotinylated goat anti-rabbit Ig pAbs (1:100, E0432, DAKO), biotinylated rabbit anti-goat Ig pAbs (1:100, E0466, DAKO), biotinylated rabbit anti-rat Ig pAbs (1:100, E0468, DAKO) or biotinylated goat anti-mouse Ig pAbs (1:100, E0433, DAKO) at room temperature for 30 min, immune complexes were visualized using Catalyzed Signal Amplification System (DAKO) according to the manufacturer's instruction. The numbers of infiltrating macrophages, T cells (CD3[+], CD4[+], and CD8[+]) and B cells into the aorta were enumerated on five randomly chosen visual fields (×400) of the sections stained with anti-F4/80, -CD3, -CD4, CD8, and -B220 Ab, respectively, and the average of the selected five fields was calculated. Histo-pathological and immunohistochemical findings were evaluated by two examiners without any prior knowledge of the experimental procedures.

**Double- or triple-color immunofluorescence analysis**. Double- or triple-color immunofluorescence analysis was conducted in order to identify the localization of CCL3, CCR5, MMP-9, TNF-α, IL-1β, NOS2, and CD206 proteins[44]. Briefly, deparaffinized sections were incubated with PBS containing 1% normal donkey serum and 1% bovine serum albumin to reduce non-specific reactions. In double-color analysis, the sections were further incubated in combinations of rabbit anti-human CCL3 pAbs (1:50, GTX52609, Gene Tex, Irvine, CA) and mouse anti-human CD68 mAb (1:100, clone PG-M1, M0876, DAKO), goat anti-human MMP-9 pAbs (1:50, sc-6840, Santa Cruz Biotechnology, Santa Cruz, CA) and mouse anti-human CD68 mAb (1:100, clone PG-M1, M0876, DAKO), goat anti-human CCR5 pAbs (1:300, GTX21673, Gene Tex, Irvine, CA) and mouse anti-human CD68 mAb (1:100, clone PG-M1, M0876, DAKO), goat anti-mouse CCL3 pAbs (1:80, GTX10381, Gene Tex, Irvine, CA) and rat anti-mouse F4/80 mAb (1:50, clone BM8, BMA Biomedicals, Switzerland), goat anti-mouse MMP-9 pAbs (1:100, sc-6840, Santa Cruz Biotechnolog) and rat anti-mouse F4/80 mAb (1:50, clone BM8, T-2028, BMA Biomedicals, Switzerland), rabbit anti-ssDNA pAbs (1:50, IBL, Gunma, Japan) and rat anti-mouse F4/80 mAb (1:50, clone BM8, T-2028, BMA Biomedicals, Switzerland), goat anti-mouse TNF-α pAbs (1:100, sc-1350, Santa Cruz Biotechnology) and rat anti-mouse F4/80 mAb (1:50, clone BM8, T-2028, BMA Biomedicals, Switzerland), or rabbit anti-mouse IL-1β pAbs (1:200, sc-7884, Santa Cruz Biotechnology), and rat anti-mouse F4/80 mAb (1:150, clone BM8, T-2028, BMA Biomedicals, Switzerland). In the triple-color analysis, sections were incubated with the combination of goat anti-mouse CCR5 pAbs (1:50, sc-6129, Santa Cruz Biotechnology), rat anti-mouse F4/80 mAb (1:50, clone BM8, T-2028, BMA Biomedicals, Switzerland), and rabbit anti-mouse NOS2 pAbs (1:200, sc-650, Santa Cruz Biotechnology) or goat anti-mouse CCR5 pAbs (1:50, sc-6129, Santa Cruz Biotechnology), rat anti-mouse F4/80 mAb (1:50, clone BM8, T-2028, BMA Biomedicals, Switzerland) and rabbit anti-mouse CD206 pAbs (1:1000, ab64693, Abcam, Cambridge, UK, 1:1000). The sections were incubated for 1 h at room temperature with the appropriate combination of following fluorochrome-conjugated secondary antibodies: Cy3-conjugated donkey anti-rat IgG pAbs (1:200, 712-165-153), Cy3-conjugated donkey anti-goat IgG pAbs (1:100, 705-165-147), Cy3-conjugated donkey anti-rabbit IgG pAbs (1:200, 711-165-152), fluorescein (FITC)-conjugated donkey anti-goat IgG pAbs (1:50, 705-095-147), FITC-conjugated donkey anti-rabbit IgG pAbs (1:50, 711-095-152), FITC-conjugated donkey anti-rat IgG pAbs (1:50, 712-095-153), and AMCA-conjugated donkey anti-goat IgG pAbs (1:50, 705-155-147) obtained from Jackson Immunoresearch Laboratories, West Grove, PA. Images were observed with a fluorescence microscope (BZ-X710 All-in-One Fluorescence Microscope, KEYENCE, Itasca, IL).

**Real-time quantitative PCR**. Total RNAs were extracted from human and mouse aortic tissues using ISOGEN (Nippon Gene, Toyama, Japan), according to the manufacturer's instructions. Total RNAs were reverse transcribed to cDNA using Prime Script Reverse Transcriptase (Takara Bio, Shiga, Japan) with oligo(dT)$_{15}$ primers. PCR was carried out using SYBR Premix Ex Taq II (Takara Bio) with specific primer sets (Takara Bio, Supplementary Tables 6 and 7)[44]. Amplification and detection of mRNA were performed using Thermal Cycler Dice Real-Time System (TP800, Takara Bio) according to the manufacturer's instructions. To normalize mRNA levels, transcript levels of *Actb* were determined in parallel for each sample, and relative transcript levels were corrected by normalization based on *Actb* transcript levels.

**MMP-9 activity assay**. Intra-aortic MMP-9 activities were determined by the Gelatin-Zymography Kit (AK47, Cosmo Bio, Tokyo, Japan) according to the manufacturer's instructions.

**Murine macrophage assay**. Peritoneal macrophages were obtained from WT, *Ccr1*[−/−], and *Ccr5*[−/−] mice 3 days after i.p. injection with 2 ml of 3% thioglycolate (Sigma-Aldrich)[63]. The resultant cells consisting of more than 95% macrophages and were suspended in antibiotic-free RPMI 1640 medium containing 10% fetal bovine serum (FBS). After the cells were incubated at 37 °C in 6 multi-well cell culture plates for 2 h, non-adherent cells were removed, and the medium was replaced. The cells were stimulated with 50 ng/ml of PMA (P1585, Sigma-Aldrich) in the presence or the absence of the indicated concentrations of murine CCL3

(450-MA-050, R&D, Minneapolis, MN) together with a p38 inhibitor (1 μM/well, SB203580, Sigma-Aldrich) or an ERK inhibitor (20 μM/well, PD98059, Sigma-Aldrich), for 24 h. In some experiments, total RNAs were extracted from murine macrophages for the determination of mRNA expression by quantitative real-time PCR (RT-PCR). In another series of experiments, the cells were homogenized with a lysis buffer (20 mM Tris-HCl (pH 7.6), 150 mM NaCl, 1% Triton X-100, 1 mM EDTA) containing Complete Protease Inhibitor Cocktail (Roche Diagnostics, Basel, Switzerland) and centrifuged at 12,000×*g* for 15 min, 4 °C to obtain lysates for Western blotting analysis[40]. The lysates (equivalent to 10 μg protein) were elec-trophoresed in a 10% SDS-PAGE and transferred onto a nylon membrane. The membrane was then incubated with the following primary Abs; rabbit anti-ERK mAb (1:1000, #4695), rabbit anti-phosphorylated (p)-ERK mAb (1:2000, #4370), rabbit anti-p38 pAbs (1:1000, #9212), rabbit anti-p-p38 mAb (1:1000, #4511), rabbit anti-JNK pAbs (1:1000, #9252), rabbit anti-p-JNK pAbs (1:1000, #9251), and rabbit anti-GAPDH mAb (1:1000, #5174) obtained from Cell Signaling, Danvers, MA. After the incubation of HRP-conjugated secondary Abs, the immune com-plexes were visualized using ECL System (BIO RAD, Hercules, CA) according to the manufacturer's instructions. Images were observed on a ChemiDoc Imaging System (BIO RAD). Band intensity was measured with ImageJ.

**Human macrophage assay**. Human THP-1 cells were obtained from American Type Culture Collection (Rockville, MD). The cells were suspended in RPMI 1640 medium containing 10% FBS and were seeded in 35-mm culture dishes at a density of $0.5 \times 10^6$ cells/ml in the presence of 50 ng/ml PMA (P1585, Sigma-Aldrich) for 24 h in a humidified atmosphere containing 5% CO to induce the dif-ferentiation into macrophages. The differentiated, adherent cells were washed with sterilized PBS (pH 7.4) and were cultured in fresh RPMI 1640 medium without PMA prior to further treatment. Macrophages were stimulated with 50 ng/ml of PMA (P1585, Sigma-Aldrich) in the presence or the absence of the indicated concentrations of human CCL3 (270-LD, R&D Systems) for the indicated time intervals. In some experiments, total RNAs were extracted for the determination of human MMP-9 mRNA expression by quantitative RT-PCR.

**Enzyme-linked immunosorbent assay**. Aortic tissues from WT mice were homogenized with PBS containing a complete protease inhibitor cocktail (Roche Diagnostics). Homogenates were centrifuged at 12,000×*g* for 15 min, 4 °C. Super-natants were used to quantify CCL3 with a commercial ELISA kit (R&D Systems), according to the manufacturer's instructions. The detection limit was 1.5 pg/ml. The total protein in the supernatant was measured with a commercial kit (BCA protein assay kit, Pierce, Rockford, IL). Data were expressed as CCL3 (pg) per total protein (mg) for each sample.

**In vivo administration of p38 MAPK inhibitor**. In a separate CaCl₂ model experiment, mice were further injected intraperitoneally with 1 mg/kg of SB239063 (a p38 inhibitor, Tocris bioscience), dissolved in 3% dimethylsulfoxide (DMSO, resolved with PBS) or vehicle only, at multiple time points, 1 h before CaCl₂ application and thereafter daily for 6 week. At the indicated time intervals after the CaCl₂ application, mice were sacrificed to measure the diameter of the aorta.

**In vivo administration of CCR5 antagonist**. In a separate CaCl₂ model experi-ment, mice were intraperitoneally injected with 10 mg/kg of maraviroc (a CCR5 antagonist, MedChem Express, Monmouth Junction, NJ) dissolved in PBS or PBS, at multiple time points, 1 h before CaCl₂ application and thereafter daily for 3 week. At the indicated time intervals after the CaCl₂ application, mice were sacrificed to measure the diameter of the aorta.

**MMP-9 inhibitor administration in vivo**. In a separate CaCl₂-model experiment, mice were intraperitoneally injected with 75 μg/mouse of an MMP-9 inhibitor (Enzo Life Science, Farmingdale, NY) dissolved in 3% DMSO and resolved with PBS or vehicle only, at multiple time points, 1 h before CaCl₂ application, and 4, 8, 12, and 16 day after CaCl₂ application. At the indicated time intervals after the CaCl₂ application, mice were sacrificed to measure the diameter of the aorta.

**Blood sample collection and flow cytometry**. Peripheral blood was collected from WT, *Ccl3*[−/−], and *Ccr5*[−/−] mice in EDTA tubes to prevent coagulation. Whole peripheral blood (200 μl) samples were stained for flow cytometry. Anti-bodies used to evaluate the cellular composition of blood were CD11b (1:100, clone M1/70, 20-0112, Tonbo Biosciences, San Diego, CA), Ly-6G (1:100, clone RB6-8C5, 35-5931, Tonbo Biosciences), F4/80 (1:80, clone BM8.1, 50-4801, Tonbo Biosciences), B220 (1:50, clone RA3-6B2, 103205, BioLegend, San Diego, CA), CD3 (1:40, clone 17A2, 75-0032, Tonbo Biosciences), CD4 (1:80, clone RM4-5, 50-0042, Tonbo Biosciences), CD8 (1:80, clone 53-6.7, 20-0081, Tonbo Biosciences), and CD45 (1:80, clone 30-F11, 103131, BioLegend). Following antibody staining, ery-throcytes were lysed using RBC Lysis Buffer (BioLegend). Samples were analyzed using a CytoFLEX S (Beckman Coulter, Brea, CA) and FlowJo software v10.5 (Becton, Dickinson and Company, Franklin Lakes, NJ). For analysis, samples were gated to first remove debris before positive gates. In addition to expression of

CD45, cellular population were further defined as CD11b$^+$Ly-6G$^+$, F4/80$^+$, B220$^+$, CD4$^+$, and CD8$^+$ cells. Isotype controls are used as negative controls.

**Human samples**. The study using human samples was approved by the Human Ethics Review Committee of Wakayama Medical University (approval No. 2253). All experiments were carried out in compliance with the declaration of Helsinki, the guidelines for ethical principles for medical research involving Human Subjects, the ethical guidelines of Wakayama Medical University, and with the relevant guidelines and regulations. Informed consent was obtained for each subject. Samples of unruptured human aneurysmal (>50 mm) aortic wall were obtained during open aortic repair procedures or autopsy cases. Aortic wall with no pathological alterations was obtained from autopsy cases. Samples were fixed with 10% neutral-buffered formalin and were then embedded with paraffin in order to prepare sections for histopathological, immunohistochemical, and immuno-fluorescence analyses as described below. In another series, samples were frozen with liquid nitrogen and stored at −80 °C until RNA extraction and quantitative RT-PCR analysis as described above.

**Statistical analysis**. Data were expressed as the mean ± SEM. For the comparison between WT and $Ccl3^{−/−}$ mice at multiple time points, two-way ANOVA followed by Dunnett's post hoc test was used. To compare the values between the two groups, a two-sided unpaired Student's $t$ test was performed. In the series of CCL3 stimulation on peritoneal macrophages in vitro, one-way ANOVA followed by Dunnett's post hoc test was used. The incidences of Ang II-induced AAA among WT, $Ccl3^{−/−}$, $Ccr1^{−/−}$, and $Ccr5^{−/−}$ mice were analyzed by the Kruskal–Wallis test followed by Steel–Dwass's test. $P < 0.05$ was considered statistically significant. All statistical analyses were performed using Statcel3 software under the supervision of a medical statistician.

**Reporting summary**. Further information on research design is available in the Nature Research Reporting Summary linked to this article.

## Data availability

The authors declare that all data are available in the article file and Supplementary information files, or available from the authors upon reasonable request. A reporting summary for this article is available as a Supplementary Information file. Source data are provided with this paper.

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

## Acknowledgements

This work was supported in part by Grants-in-Aid for Scientific Research (A) (grant 20249040 and grant 25253055 to T.K.), for Young Scientists (A) (grant 20689015 to Y.I.) an for Scientific Research (B) (grant 20H03957 to Y.I.) from Japan Society for the Promotion of Science, by Extramural Collaborative Research Grant of Cancer Research Institute, Kanazawa University (to Y.I) and by Research Grant on Priority Areas (to T.K.) from Wakayama Medical University.

## Author contributions

Y.I. and T.K. formulated the hypothesis and designed the project; Y.I. performed the main experiments; A.K. provided the technical assistance and discussion; Y.K., M.N., A.T., and M.F. helped with the some experimental procedures; N.M. and T.K. oversaw the experiments and provided the main funding for the project; Y.I., N.M., and T.K. participated in writing the paper.

## Competing interests

The authors declare no competing interests.
