## [Peer Review File · Nature Communications]

Reviewers' comments:

Reviewer #1 (Remarks to the Author):

Kondo et al investigated the role of immune cell chemotaxis regulators, chemokine CCL3 and its receptor CCR5 in the development of experimental murine abdominal aortic disease. The recruitment of monocytes to the aneurysmal tissue is well studied and documented in the current literature (Raffort et al, *Nat Reviews cardiology*, 2017). The unexpected protective finding of CCL3 and CCR5-loss-of function in aneurysms disease would be in fact expected based on direct similar evidence provided by a selection of some of the published work:

1- In the similar CaCl₂ model, MacTaggart et al (*Surgery*, 2007) demonstrated that the deletion of CCR5 did not protect against disease compared to CCR2 deletion.

2- Hoh et al (*Circulation*, 2011), showed that MCP-1 induce aortic repair via CCL3 (MIP1a) dependent pathway in aneurysms. They demonstrated that CCL3 increased recruitment of fibroblast and macrophages and promoted aortic repair.

3- Dobaczewski et al (*Am J Pathol*, 2010) demonstrated that the deletion of CCR5 promotes inflammation and defective vascular remodeling via the impaired recruitment of regulatory foxp3 T cells.

4- Iida, Y. et al demonstrated that inhibition of CXCL4-CCL5 heterodimer refrained experimental aortic aneurysm initiation and progression.

Arterioscler. Thromb. Vasc. Biol. 33, 718–726 (2013)

Although it is appreciated that the work by Kondo et al utilized various methodologies of depletion and supplementation of recombinant CCL3 to test its impact on AAA, the work lacks originality and novelty, is mostly descriptive and the mechanisms underlying the protective role of CCL3/CCR5 is insufficiently developed.

Major concerns:

- The authors did not explore the possibility of compensatory feedback mechanisms of other chemokines in the single knock out model. Possibilities of double knock-out and Lox/Cre macrophage specific loss of function of CCL3 and CCR5. This would be key in order to understand fine-tuned mechanisms given their findings.
- It is important to validate all their findings in another well-established murine AAA modeled by the exposition of mice to continuous infusion of Angiotensin II, as the inflammatory regulators are more dynamic and elastin damage more evident in this model. Modest elastin damage shown in WT in Figure 1G.
- In Figure 1c, CCL3 levels increased abruptly following CaCl₂ injury and gradually declined until 6 weeks post disease induction. It would be really novel and interesting to investigate what are the factors that causes the decline of CCL3, which would provide key insights into its protective role in aneurysms.
- Figure 1D is confusing. Where in the aorta are CCL3 positive macrophages localized? No orientation is provided.
- The authors did not profile the whole immune cell behavior in the whole body knock out of CCL3 and CCR5. Other immune cells that have been demonstrated to play critical roles in aneurysms disease such as T and B cells which also express these chemokines. Investigating the recruitment of CD3 positive cells is insufficient in this context. In Figure 2a, the authors demonstrate a trend in increased CD3+ lymphocytes in the aortic tissue. Are these cells contributing to the protective role of CCL3?
- In Figure 2b, the authors demonstrate increase of F4/80 positive macrophages in CCL3 knock out mice. They show that this is associated with MMP9 expression, but direct proof is not provided. Importantly the activity of MMP9 is not analyzed. Effect on inhibitors of MMP (TIMP) are not tested. Effect of CCL3 on macrophage capacity to regulate matrix production is not addressed. What is the origin and phenotype of CCR5 monocytes/macrophages? Diverse macrophages phenotypes, pro- and anti-inflammatory, has been suggested to play a role in AAA development. What subtype are

these macrophages? This should be comprehensively characterized by flow cytometry. It is likely that they are subpopulations of refrained repair capacities. This should be characterized both in circulating monocytes and in transmural tissue macrophages. In Figure 2O, MMP9 seems to be expressed by other non-F4/80 positive cells, consistent with its release by vascular smooth muscle cells. The impact of the absence on vascular smooth muscle cell death, proteolytic phenotype is not addressed.

- A comprehensive profiling of the protective role of CCL3 by RNA sequencing or single cell RNA sequencing (other Omics approach) would provide key novel protective mechanisms of CCL3 on extracellular matrix degradation. Exploring the role of MAPK/ERK axis in inducing MMP9 has been extensively studied in the field of aneurysm disease. The mechanism is poorly developed in the current state of this work. The rationale for focusing on the role of MMP9 is unclear. Importantly, does CCL3-mediated signaling regulate other MMPs (MMP12-macrophage specific MMP)? Sequencing analysis would be helpful to address this.

Minor comments:

1-The introduction of the manuscript stating that "Abdominal aortic aneurysm (AAA) results from atherosclerotic changes..." is an overstatement of the actual etiology of the disease, alluding that all AAA arise only from atherosclerotic lesions. Appropriate natural history of AAA should be stated as is.

2-In figure 1H, elastin break is quantified at 20 weeks. Mice are harvested at 6 weeks throughout the manuscript? Why was Cx3Cr1-/- investigated here. Is this a typo mistake?

Reviewer #2 (Remarks to the Author):

One of the issues with this or any model of AAA is that none has translated to an effective therapy. So the question becomes whether they are useful in understanding human disease or only relevant to mice. One approach that has been suggesting is adding a second model but I don't believe this gets any closer to knowing if the data are clinically relevant. The value of a well-characterized model is that it may identify important biologic mechanisms about molecules that play important roles in human disease. As such, I think the data may be very important.

1) The data quality generally appears high except for the zymography. This is critical since changes in MMP-9 levels are the key finding of the paper. Zymography provides a reliable and exquisitely sensitive measure of latent and active MMP-9. This is important to this study since the gene expression data are not clean. Image N in figure 1 is hard to interpret and poor quality. Latent and active forms should be identifiable. Figure 2F also raises a question since the MMP-9 expression at 1 week in CaCl₂ treated mice with IgG infusion show essentially no MMP-9 expression. Supporting, quality zymograms are needed to verify protein changes in MMP-9

2) The changes identified may be relevant and important to humans but doing the work in another model will not address this. Rather, validating the MMP-9 suppression by CCL3 in human, in addition to murine, macrophages would extend the findings beyond this mouse model.

Minor change- the background work on the model suggests it was developed in rodents (ref 4-5). The model was actually developed in rabbit carotid arteries.

Reviewer #3 (Remarks to the Author):

General Comments

The manuscript by Ishida, et. al., dealing with effects of ccl3 and mmp3 on the abdominal aortic aneurysm (AAA) induced by injection of CaCl₂ in mice is of some interest. The authors use ccr3 -/- and ccr5 -/- mice as well as recombinant murine ccl3 and anti-ccl3 to suggest the involvement of increased ccl3 in inducing mmp3 as a means to dampen the AAA induced by CaCl₂, effects

mediated in large part through one of the ccl3 receptors, ccr5. The data as presented seems reasonable and the experiments as far as they have gone are strong. This is not an uncomplicated study. It may be a bit confusing to the readers who have little or no knowledge about the redundancy between the chemokine ligands and their chemokine receptors. This can be helped by making it clear about these chemokine ligand-receptor interactions and providing a Figure at the end that clearly diagrams what has been done and the interpretations given. However, a main problem is that the conclusion that ccl3-ccr5 interactions influence mmp3 production which itself is the main modifying agent on Cacl2-AAA defects is based mainly on associations provided in the paper and the work of others that have linked mmp3 with counteracting AAA effects. The last paragraph of the Results section tries to tie this together by showing the in vivo effects of ccl3 or mmp3 production by macrophages stimulated by the phorbol ester, PMA, and used this system to implicate MAPK signaling pathways and p38 MAPK. One problem here is that PMA by-passes receptor mediated effects and is not likely to mimic in vivo conditions of such ccl3-ccr5-mmp interactions. Perhaps additional studies using dual mmp3 $-/-$ and chemokine receptor $-/-$ mice might provide more relevant information.

Specific Comments

1. Make it clear that ccl3 can bind to three known chemokine receptors (ccr3, ccr5, and ccr1) so that the readers are more aware of why you look at these chemokine receptor $-/-$ mice. You can decide where this information best fits.
2. What about effects on dual ccr5 $-/-$ and ccr3 $-/-$ mice, and dual chemokine receptor $-/-$ mice and mmp3 $-/-$ mice? These types of experiments might provide more direct data than now present for your conclusions so that you don't have to rely only on your work which at present only suggests association, and the published work of others that provide evidence for mmp3 production and modification of AAA-induced effects.
3. There must be another way to demonstrate ccl3 effects on cell production of mmp3 than by using PMA stimulate production of mmp3, which is far from a physiological effect even though it has been used by others. To link p38 MAPK, can you use an inhibitor like SB203580 in vivo your mouse model of CaCl₂ induced AAA, along with appropriate controls that can demonstrate the specificity of the p38 MAPK inhibitor?

Point-by-point reply

Reviewer #1

Kondo et al investigated the role of immune cell chemotaxis regulators, chemokine CCL3 and its receptor CCR5 in the development of experimental murine abdominal aortic disease. The recruitment of monocytes to the aneurysmal tissue is well studied and documented in the current literature (Raffort et al, Nat Reviews cardiology, 2017). The unexpected protective finding of CCL3 and CCR5-loss-of-function in aneurysms disease would be in fact expected based on direct similar evidence provided by a selection of some of the published work:

1- *In the similar CaCl₂ model, MacTaggart et al (Surgery, 2007) demonstrated that the deletion of CCR5 did not protect against disease compared to CCR2 deletion.*

In MacTaggart's study, the absence of CCR5 had no effects on the development of CaCl₂-induced aortic aneurysm. Their observations were not consistent with our results using the same model. There was a difference in the genetic background of the used mice strain; SV129 mice in MacTaggart's study and C57BL/6 in ours. The difference of genetic background in mice would account for the discrepant results. In the present study, we demonstrated that the absence of CCR5 exaggerated the development of AAA. As we discussed in the third paragraph of page 16, in line with our observations, the previous studies revealed that the genetic deletion of CCR5 in humans increased the incidence of AAA (ref 61 and 62). We think that these clinical studies may rather favor our claim that CCR5 had protective roles in the development of AAA.

2- *Hoh et al (Circulation, 2011), showed that MCP-1 induce aortic repair via CCL3 (MIP1a) dependent pathway in aneurysms. They demonstrated that CCL3 increased recruitment of fibroblast and macrophages and promoted aortic repair.*

We are afraid that it is not appropriate to directly extrapolate Hoh's ones into our CaCl₂-induced AAA model because Hoh employed carotid artery aneurysms induced by porcine pancreatic elastase, where different mechanism could be involved.

3- *Dobaczewski et al (Am J Pathol, 2010) demonstrated that the deletion of CCR5 promotes inflammation and defective vascular remodeling via the impaired recruitment of regulatory foxp3 T cells.*

We are afraid that it is not appropriate to directly compare our results with Dobaczewski's ones, because there is a difference in the aneurysm model between the two studies. Dobaczewski examined the roles of CCR5 in tissue remodeling after myocardial infarction, suggesting that the recruitment of CCR5⁺ regulatory T cells played important roles in suppressing inflammatory responses, and eventually promoted damaged-tissue remodeling. On the contrary, in this model, we demonstrated that CD4⁺ T cells were

increased to a similar extent in WT and CCL3^{-/-} mice after CaCl₂ treatment (Supplemental Figure 4). Thus, it may be a remote possibility that Treg cells could contribute to enhanced AAA formation in CCL3^{-/-} mice.

4- Iida, Y. et al demonstrated that inhibition of CXCL4-CCL5 heterodimer refrained experimental aortic aneurysm initiation and progression.

The aneurysm model (elastase infusion model) in Iida's study is different from ours. Moreover, we examined the expression of CXCL4 and CCL5, and found no differences in CXCL4 and CCL5 expression between WT and CCL3^{-/-} mice (for only review).

Accordingly, we modified text in the revision with the citation of these references 24 to 26 (page 4, paragraph 3; page 12, paragraph 1).

Additionally, we observed that CaCl₂ application induced Ccl3^{-/-} mice to exhibit enhanced intra-aortic gene expression of Ccl2 (a potent macrophage tropic chemokine) to a larger extent than WT mice. Although CCL2 was expressed in macrophages, our in vitro study revealed that CCL3 had no effects on Ccl2 gene expression in macrophages. Thus, we assumed that the enhanced CCL2 expression in Ccl3^{-/-} mice could be ascribed to enhanced recruitment of macrophage in Ccl3^{-/-} mice. Accordingly, we demonstrated these results as Fig. S8, and modified the section of Discussion in the revision (page 15, paragraph 2)

Although it is appreciated that the work by Kondo et al utilized various methodologies of depletion and supplementation of recombinant CCL3 to test its impact on AAA, the work is lacks originality and novelty, is mostly descriptive and the mechanisms underlying the protective role of CCL3/CCR5 is insufficiently developed.

In the original manuscript, we proved that CCL3, a characteristic pro-inflammatory chemokine, can be protective for CaCl₂-induced AAA formation. In order to validate the generality of the observations, we further revealed that CCL3 can be protective for another AAA model, angiotensin II (AngII)-induced AAA formation. We believe that these observations have provided the first definitive evidence to indicate the protective role of CCL3 for AAA formation and therefore, has originality and novelty.

In order to clarify the molecular mechanisms, we first tried to elucidate the roles of two

CCL3-specific receptors, CCR1 and CCR5. We identified CCR5 but not CCR1 as the responsible receptor for CCL3 activities in these two models. In the original manuscript, we observed that MMP-9 expression was enhanced in AAA sites in *Ccl3*^{-/-} mice. Moreover, based on in vitro observations, we proposed that CCL3 could inhibit the expression of MMP-9, an enzyme crucially involved in aortic formation, by inducing p38 MAPK phosphorylation. We admitted that these in vitro observations were just circumstantial evidence and therefore, we examined the effects of a MMP-9 inhibitor and a p38 MAPK inhibitor on CaCl₂-induced AAA formation. Consistent with *in vitro* observations, a MMP-9 inhibitor reduced AAA formation (Fig. 4i) whereas a p38 MAPK inhibitor aggravated AAA formation (Fig. 7h). We believe that these observations have provided a mechanistic insight on the protective roles of the CCL3-CCR5 axis for CaCl₂-induced AAA formation.

Accordingly, we modified the sections of Results (page 9, paragraph 2, page 11, paragraphs 1 and 2) and Discussion (page 13, paragraph 3, page 15, paragraphs 1 and 2)

Major concerns:

- *The authors did not explore the possibility of compensatory feedback mechanisms of other chemokines in the single knock out model. Possibilities of double knock-out and Lox/Cre macrophage specific loss of function of CCL3 and CCR5. This would be key in order to understand fine-tuned mechanisms given their findings.*

Since double knock-out or Lox/Cre macrophage specific loss of function of CCL3 and CCR5 mice were not available, we administered a CCR5 antagonist, maraviroc, to *Ccl3*^{-/-} mice in order to doubly block CCL3 and CCR5. Expectedly, maraviroc-treated WT mice exhibited exaggerated CaCl₂-induced aneurysm formation, compared with vehicle-treated WT mice similarly as observed in *Ccr5*^{-/-} mice. On the contrary, there was no significant difference in the aortic diameters between maraviroc-treated and vehicle-treated *Ccl3*^{-/-} mice after CaCl₂ application (Fig. 4g), indicating no compensatory effects of other CCR5 ligands through CCR5. Supporting this notion, the absence of CCR1, another receptor for CCL3, had no influence on CaCl₂-induced aneurysm formation as observed in CaCl₂-treated WT mice. Moreover, CCL3 *in vitro* suppressed MMP-9 expression in macrophages derived from WT or *Ccr1*^{-/-} mice but not *Ccr5*^{-/-} mice (Fig. 7b). Thus, we believe that these results can exclude the possibility of compensatory feedback mechanisms of other chemokines. Hence, we modified the section of Results in the revised version (page 8, paragraph 3 to page 9, paragraph 1).

- *It is important to validate all their findings in another well-established murine AAA modeled by the exposition of mice to continuous infusion of Angiotensin II, as the inflammatory regulators are more dynamic and elastin damage more evident in this model. Modest elastin damage shown in WT in Figure 1G.*

In response to the recommendation, we consecutively infused angiotensin II (AngII) to

WT, *Ccl3*^{-/-}, *Ccr1*^{-/-}, and *Ccr5*^{-/-} mice. AngII treatment elevated blood pressure to a similar extent in these four strains (Fig. 8a). As shown in Fig. 8b to d, AngII application exaggerated aneurysm formation in *Ccl3*^{-/-} and *Ccr5*^{-/-} mice but not *Ccr1*^{-/-} mice, compared with WT mice 28 days after the treatment. Thus, the CCL3-CCR5 axis has protective roles in both CaCl₂- and AngII-induced aneurysm formation. We modified the section of Results (page 11, paragraph 2), Discussion (page 16, paragraph 3) and Methods (page 19, paragraph 2, 3) in the revised version.

- In Figure 1c, CCL3 levels increased abruptly following Cacl2 injury and gradually declined until 6 weeks post disease induction. It would be really novel and interesting to investigate what are the factors that causes the decline of CCL3, which would provide key insights into its protective role in aneurysms.

Since we assumed that macrophages were the major source of CCL3 in CaCl₂-induced aneurysm model in WT mice, we examined the time kinetic changes in macrophage recruitment into aorta. CCL3 expression in aorta correlated with macrophage recruitment. We observed that the number of apoptotic macrophages increased with the reciprocal decrease in CCL3 expression, indicating the association of apoptosis in macrophages with declined CCL3 expression (Fig. S2). Hence, we modified the sections of Results (page 6, paragraph 1) in the revision.

- Figure 1D is confusing. Where in the aorta are CCL3 positive macrophages localized? No orientation is provided.

In response to the comment, we replaced the new Fig. 1d (low-power and high-power magnification) with the old one in the revised version

- The authors did not profile the whole immune cell behavior in the whole body knock out of CCL3 and CCR5. Other immune cells that have been demonstrated to play critical roles in aneurysms disease such as T and B cells which also express these chemokines. Investigating the recruitment of CD3 positive cells is insufficient in this context. In Figure 2a, the authors demonstrate a trend in increased CD3+ lymphocytes in the aortic tissue. Are these cells contributing to the protective role of CCL3?

In response to the comment, we examined the changes in circulating immune cells in WT, *Ccl3*^{-/-} and *Ccr5*^{-/-} mice after CaCl₂ treatment and observed no significant differences in the changes among these strains as shown in Fig. S3. We modified the sections of Result (page 7, paragraph 2) and Methods (page 24, paragraph 3).

Moreover, we investigated the numbers of CD3⁺, CD4⁺, CD8⁺, and B220⁺ cells in aortic aneurysm in WT and *Ccl3*^{-/-} mice and detected no significant differences in CD3⁺, CD4⁺ cells, CD8⁺ T cells and B220⁺ B cells between WT and *Ccl3*^{-/-} mice as shown in Fig. S4. Thus, it may be unlikely that CD3⁺ T cell infiltration was responsible for enhanced CCL3

expression in AAA formation. Moreover, given the double-color immunofluorescence analysis (Fig. 1d), it may be reasonable to conclude that macrophages were the main cellular source of CCL3. Hence, we modified the Result (page 7, paragraph 2) and Methods (page 21, paragraph 1).

- In Figure 2b, the authors demonstrate increase of F4/80 positive macrophages in CCL3 knock out mice. They show that this is associated with MMP9 expression, but direct proof is not provided. Importantly the activity of MMP9 is not analyzed. Effect on inhibitors of MMP (TIMP) are not tested. Effect of CCL3 on macrophage capacity to regulate matrix production is not addressed. What is the origin and phenotype of CCR5 monocytes/macrophages? Diverse macrophages phenotypes, pro- and anti-inflammatory, has been suggested to play a role in AAA development. What subtype are these macrophages? This should be comprehensively characterized by flow cytometry. It is likely that they are subpopulations of refrained repair capacities. This should be characterized both in circulating monocytes and in transmural tissue macrophages. In Figure 2O, MMP9 seems to be expressed by other non-F4/80 positive cells, consistent with its release by vascular smooth muscle cells. The impact of the absence on vascular smooth muscle cell death, proteolytic phenotype is not addressed.

We demonstrated that *Ccl3*^{-/-} mice exhibited exaggerated macrophage and neutrophil recruitment with enhanced MMP-9 expression. Given the crucial involvement of MMP-9 in macrophage and neutrophil infiltration (ref. 50, 51), it is reasonable to assume that augmented MMP-9 expression can account for macrophage and neutrophil infiltration in *Ccl3*^{-/-} mice.

In response to the comments, we examined the intra-aortic MMP-9 activity using gelatin zymography and observed that both pro and active forms of MMP-9 were markedly increased in *Ccl3*^{-/-} mice and anti-CCL3-treated WT mice as shown in Fig. 1n, Fig. 2g. Moreover, we examined the effects of a MMP-9 inhibitor on aneurysm formation and demonstrated that a MMP-9 inhibitor alleviated CaCl₂-induced aneurysm formation in *Ccl3*^{-/-} and *Ccr5*^{-/-} mice as shown in Fig. 4i. These observations would indicate that enhanced aneurysm formation in *Ccl3*^{-/-} and *Ccr5*^{-/-} mice was ascribed to augmented MMP-9 activity. Hence, we modified the section of Results (page 9, paragraph 2), Discussion (page 15, paragraph 2), and Methods (page 24, paragraph 3).

Furthermore, we examined the effects of CCL3 on MMP-9 expression in mouse macrophages and demonstrated that CCL3 suppressed MMP-9 gene expression in macrophages obtained from WT mice in a dose-dependent manner. However, these effects were cancelled in *Ccr5*^{-/-} macrophages but not *Ccr1*^{-/-} macrophages as shown in Fig. 7, a and b. Moreover, CCL3 also inhibited MMP-9 gene expression in a human macrophage cell line, THP-1 cells as shown in Fig. S6. Thus, we think that we have provided definitive evidence to indicate that CCL3 can depress MMP-9 expression in macrophages by acting its specific receptor, CCR5. Hence, we modified the section of Results (page 10, paragraph 3) and Methods (page 23, paragraph 2).

Macrophage polarization, particularly a higher M1/M2 ratio, was associated with AAA formation (ref. 29, 31). Thus, in response to the comment, we examined macrophage polarization in WT and *Ccl3*^{-/-} mice after CaCl₂ treatment. CaCl₂ application up-regulated the intra-aortic gene expression of *Nos2* (M1 marker) and *Cd206* (M2 marker) in WT mice. The absence of CCL3 further enhanced *Nos2* gene expression (Fig. 5a) but reduced *Cd206* mRNA expression, compared with WT ones (Fig. 5b). Immunohistochemical analysis further demonstrated a higher ration of NOS2⁺ M1- to CD206⁺ M2-macrophages in *Ccl3*^{-/-} mice than WT ones (Fig. 5c and d). Although CCR5 was expressed on both M1 and M2 macrophages (Fig. 5c and e), these observations implied the CCL3 deficiency could favor macrophage polarization into M1-like phenotype, thereby contributing to AAA formation. Hence, we modified the section of Results (page 9, paragraph 3) and Discussion (page 15, paragraph 3).

We could not completely exclude the possibility of MMP-9 expression in vascular smooth muscle cells as the reviewer pointed out. However, the analysis using bone chimeric mice revealed that aneurysm formation and MMP-9 expression were enhanced by CCL3 deficiency restricted to bone marrow-derived cells. Moreover, consistent with the previous study that MMP-9 was predominantly derived from macrophage in the development of AAA (ref. 45), our double-color immunofluorescence analysis identified macrophages as a major MMP-9-expressing cells. We think that these observations would favor the notion that macrophages were a major source of MMP-9. Accordingly, we modified the Discussion section (page 13, paragraph 2).

- A comprehensive profiling of the protective role of CCL3 by RNA sequencing or single cell RNA sequencing (other Omics approach) would provide key novel protective mechanisms of CCL3 on extracellular matrix degradation. Exploring the role of MAPK/ERK axis in inducing MMP9 has been extensively studied in the field of aneurysm disease. The mechanism is poorly developed in the current state of this work. The rationale for focusing on the role of MMP9 is unclear. Importantly, does CCL3-mediated signaling regulate other MMPs (MMP12-macrophage specific MMP)? Sequencing analysis would be helpful to address this.

In response to the comment, we examined the gene expression of other MMPs and TIMPs, which are presumed to be involved in aneurysm formation. We observed no significant differences in *Timp1*, *Timp2*, *Mmp2*, *Mmp3*, *Mmp10*, and *Mmp12* mRNA expression between WT and *Ccl3*^{-/-} mice at 1 w after CaCl₂ treatment (Figure S5). Thus, we believe that it is likely that augmented AAA formation in *Ccl3*^{-/-} or *Ccr5*^{-/-} mice can be ascribed to enhanced MMP-9 activities. Hence, we modified the section of Results (page 7, paragraph 2).

Minor comments:

1- *The introduction of the manuscript stating that “Abdominal aortic aneurysm (AAA) results from atherosclerotic changes...” is an overstatement of the actual etiology of the disease, alluding that all AAA arise only from atherosclerotic lesions. Appropriate natural history of AAA should be stated as is.*

In accordance with comment, we modified the section of Introduction (page 3, paragraph 1) in the revision.

2- *In figure 1H, elastin break is quantified at 20 weeks. Mice are harvested at 6 weeks throughout the manuscript? Why was Cx3Cr1-/- investigated here. Is this a typo mistake?*

We appreciate the comment and corrected the typo mistake in the legend to Figure 1h in the revised manuscript.

Reviewer #2

One of the issues with this or any model of AAA is that none has translated to an effective therapy. So the question becomes whether they are useful in understanding human disease or only relevant to mice. One approach that has been suggesting is adding a second model but I don't believe this gets any closer to knowing if the data are clinically relevant. The value of a well-characterized model is that it may identify important biologic mechanisms about molecules that play important roles in human disease. As such, I think the data may be very important.

In response to the recommendation, we consecutively infused angiotensin II (AngII) to WT, *Ccl3*^{-/-}, *Ccr1*^{-/-}, and *Ccr5*^{-/-} mice. AngII treatment elevated blood pressure to a similar extent in these four strains (Fig. 8a). As shown in Fig. 8b to d, AngII application exaggerated aneurysm formation in *Ccl3*^{-/-} and *Ccr5*^{-/-} mice but not *Ccr1*^{-/-} mice, compared with WT mice 28 days after the treatment. Thus, the CCL3-CCR5 axis has protective roles in both CaCl₂- and AngII-induced aneurysm formation. We modified the section of Results (page 11, paragraph 2), Discussion (page 16, paragraph 3) and Methods (page 19, paragraph 2, 3) in the revised version.

Moreover, we discussed the clinical relevance of the present observations, by mentioning the increased incidence of abdominal aneurysm in individual with CCR5Δ32 allele, which can lead to depressed CCR5 expression similarly as *Ccr5*^{-/-} mice (page 16, paragraph 3).

1) *The data quality generally appears high except for the zymography. This is critical since changes in MMP-9 levels are the key finding of the paper. Zymography provides a reliable and exquisitely sensitive measure of latent and active MMP-9. This is important to this study since the gene expression data are not clean. Image N in figure 1 is hard to interpret and poor quality. Latent and active forms should be identifiable. Figure 2F also*

raises a question since the MMP-9 expression at 1 week in CaCl₂ treated mice with IgG infusion show essentially no MMP-9 expression. Supporting, quality zymograms are needed to verify protein changes in MMP-9.

In response to the comments, we performed gelatin zymography again in the revision process and replaced the new Fig. 1n and Fig. 2g with the old ones, in the revised manuscript.

2) The changes identified may be relevant and important to humans but doing the work in another model will not address this. Rather, validating the MMP-9 suppression by CCL3 in human, in addition to murine, macrophages would extend the findings beyond this mouse model.

In response to the comments, we examined the effects of CCL3 on MMP-9 expression in THP-1 cells (a human macrophage cell line). Consistent with the results of *in vitro* study using murine macrophages, CCL3 suppressed PMA-induced *Mmp-9* gene expression in THP-1 cells (Fig. S6). Hence, we modified the section of Results (page 10, paragraph 3) and Methods (page 23, paragraph 2) in the revised version.

Additionally, we conducted immunohistochemical analyses on human AAA, which showed the presence of a large number of CD68⁺ macrophages, MMP-9⁺ cells, CCL3⁺ cells and CCR5⁺ cells in aortic aneurysm lesion (Fig. S1a and b). A double-color immunofluorescence analysis further demonstrated that MMP-9, CCL3, and CCR5 were expressed on CD68⁺ macrophages in human aortic aneurysm tissues (Fig. S1c). Moreover, we observed that *Mmp9* mRNA expression was significantly increased in AAA samples, compared with controls (Fig. S1d). We hope that these observations would indicate that the results arising from the animal experiments are relevant to human aneurysm formation processes.

Minor change- *the background work on the model suggests it was developed in rodents (ref 4-5). The model was actually developed in rabbit carotid arteries.*

In response to the comment, we modified the text (page 3, paragraph 2) in the revision.

Reviewer #3

General Comments

The manuscript by Ishida, et. al., dealing with effects of ccl3 and mmp3 on the abdominal aortic aneurysm (AAA) induced by injection of CaCl₂ in mice is of some interest. The authors use ccr3 -/- and ccr5 -/- mice as well as recombinant murine ccl3 and anti-ccl3 to suggest the involvement of increased ccl3 in inducing mmp3 as a means to dampen the AAA induced by CaCl₂, effects mediated in large part through one of the ccl3 receptors, ccr5. The data as presented seems reasonable and the experiments as far as they have

*gone are strong. This is not an uncomplicated study. It may be a bit confusing to the readers who have little or no knowledge about the redundancy between the chemokine ligands and their chemokine receptors. This can be helped by making it clear about these chemokine ligand-receptor interactions and providing a Figure at the end that clearly diagrams what has been done and the interpretations given. However, a main problem is that the conclusion that *ccl3-ccr5* interactions influence *mmp3* production which itself is the main modifying agent on *Cacl2-AAA* defects is based mainly on associations provided in the paper and the work of others that have linked *mmp3* with counteracting AAA effects. The last paragraph of the Results section tries to tie this together by showing the *in vivo* effects of *ccl3* or *mmp3* production by macrophages stimulated by the phorbol ester, PMA, and used this system to implicate MAPK signaling pathways and p38 MAPK. One problem here is that PMA by-passes receptor mediated effects and is not likely to mimic *in vivo* conditions of such *ccl3-ccr5-mmp* interactions. Perhaps additional studies using dual *mmp3* *-/-* and chemokine receptor *-/-* mice might provide more relevant information.*

We appreciate your comments. In order to enhance the readability, we made the new Figure S9, which illustrate the roles of the CCL3-CCR5 interaction in aortic aneurysm formation.

In response to the recommendation to elucidate the roles of MMP-9 in augmented AAA formation in *Ccl3*^{-/-} and *Ccr5*^{-/-} mice, we treated these mice in order to mimic double knockout mice of MMP-9 and CCL3 or CCR5 as we described in the response to the specific comment 2.

Specific Comments

1. *Make it clear that *ccl3* can bind to three known chemokine receptors (*ccr3*, *ccr5*, and *ccr1*) so that the readers are more aware of why you look at these chemokine receptor *-/-* mice. You can decide where this information best fits.*

Even in the original paper (www.jbc.org/content/270/28/16491.long) which reported the cloning of CCR3, CCR3 can bind to CCL3 with a markedly lower affinity than CCR1 or CCR5, suggesting that CCR3 was not a physiologically specific receptor for CCL3. Thus, we used *Ccr1*^{-/-} and *Ccr5*^{-/-} mice in the present study.

2. *What about effects on dual *ccr5* *-/-* and *ccr3* *-/-* mice, and dual chemokine receptor *-/-* mice and *mmp3* *-/-* mice? These types of experiments might provide more direct data than now present for your conclusions so that you don't have to rely only on your work which at present only suggests association, and the published work of others that provide evidence for *mmp3* production and modification of AAA-induced effects.*

In the present study, we examined the gene expression of MMPs and TIMPs which have been reported to be involved in aneurysm formation. We observed exaggerated MMP-9 expression in *Ccl3*^{-/-} mice, compared with WT mice, but detected no significant

differences in Timp1, Timp2, Mmp2, Mmp3, Mmp10, and Mmp12 between WT and *Ccl3*^{-/-} mice at 1 w after CaCl₂ treatment (Fig. S5). Thus, we dared not examine the roles of MMP-3 in this model. As we described in the response to the previous comment, it is very unlikely that CCR3 has a role in CCL3-mediated pathological changes and therefore, we dared not examine the role of CCR3 in this process. In contrast, in order to mimic double knockout mice of MMP-9 and CCL3 or CCR5, we injected a MMP-9 inhibitor into *Ccl3*^{-/-} and *Ccr5*^{-/-} mice and observed that the treatment significantly reduced CaCl₂-induced AAA formation in *Ccl3*^{-/-} and *Ccr5*^{-/-} mice (Fig. 4i). These observations implied that enhanced MMP-9 expression was responsible for enhanced AAA formation in *Ccl3*^{-/-} and *Ccr5*^{-/-} mice. Collectively, we modified the section of Results (page 9, paragraph 2) and Methods (page 24, paragraph 3) in the revised version.

3. *There must be another way to demonstrate ccl3 effects on cell production of mmp3 than by using PMA stimulate production of mmp3, which is far from a physiological effect even though it has been used by others. To link p38 MAPK, can you use an inhibitor like SB203580 in vivo your mouse model of CaCl2 induced AAA, along with appropriate controls that can demonstrate the specificity of the p38 MAPK inhibitor?*

In response to the recommendation, we examined the in vivo effects of a p38 MAPK inhibitor (SB239063) on CaCl₂-induced aneurysm and observed that SB239063 exaggerated CaCl₂-induced AAA formation in WT mice as shown in Fig. 7h. Collectively, we modified the section of Results (page 11, paragraph 1) and Methods (page 23, paragraph 4) in the revised version.

Reviewers' comments:

Reviewer #1 (Remarks to the Author):

The authors addressed most of my concerns.

However, some issues persist.

Major point:

1- Experiment with angiotensin II infusion is questionable. Typically, mice deficient for apolipoprotein E respond better to Ang II treatment at dosage used and develop AAA in 80% of cases. WT, C57/B6 mice respond less efficiently but the incidence of AAA can be increased in the presence of PCSK9 overexpression.

What age and sex of WT, CCR5^{-/-} and CCL3^{-/-} mice used for the AngII experiment?

What was the incidence of AAA? Also the size of the aorta reported in figure 8c does not seem to match images shown in the left-the aneurysms seems to be bigger than 1000 μ m. Are the authors measuring diameter of aorta or radius? It is unlikely that the size of the aorta to be 500 μ m? Did they use Doppler for measurements?

2-The elastin staining is not clear as is. Higher magnification will better depict elastin breaks.

3-Did MMP9 levels change in angiotensin II model?

Minor points:

1-The authors did not change the statement that "Abdominal aortic aneurysm (AAA) results from atherosclerotic changes...". This sentence is still in the second paragraph of the introduction. As I commented before, it is wrong and misleading to state that all AAA arise from atherosclerotic lesions.

2-The simplicity of the schematic illustrating their findings in the last supplementary figure needs improvement. It is confusing and unclear as is.

Reviewer #2 (Remarks to the Author):

My concerns were related to quality of data on the MMPs and relevance of the models to human AAA. The authors have added new data regarding the MMPs, extended and confirmed similar results in a second animal model and analyzed some human samples. This final analysis confirmed expression of CCR3 and 5 in macrophages in human AAA tissue.

Point-by-point reply

Reviewer #1

Major points

1. *Experiment with angiotensin II infusion is questionable. Typically, mice deficient for apolipoprotein E respond better to Ang II treatment at dosage used and develop AAA in 80% of cases. WT, C57BL/6 mice respond less efficiently but the incidence of AAA can be increased in the presence of PCK9 overexpression.*

What age and sex of WT, CCR5^{-/-} and CCL3^{-/-} mice used for the Ang II experiment?

What was the incidence of AAA? Also the size of the aorta reported in figure 8c dose not seem to match images shown in the left-the aneurysms seems to be bigger than 1000 μm . Are the authors measuring diameter of aorta or radius? It is unlikely that the size of the aorta to be 500 μm ? Did they use Doppler for measurements?

In all animal experiments including Ang II-induced AAA model of the present study, we used 8-week-old male mice in each strain as described in Materials and Methods section.

In response to the comments, we performed additional analyses on Ang II-infused aneurysm model in accordance with the previous study (Ref. No. 33). Namely, we examined the suprarenal aorta in vivo by ultrasound imaging at 4 weeks after Ang II infusion, and obtained the suprarenal aortic samples which were subjected to the measurement of their external diameter by an independent researcher *ex vivo* under a microscope. Moreover, we employed the following condition as the criteria for Ang II-induced AAA formation: more than 30 % increase in the external diameters of the suprarenal aorta, compared with the average of control mice. Since the average of the external diameters of the suprarenal aortic width was 765 μm in control mice, we judged that AAA developed when the external diameters were more than 995 μm . Additionally, we examined intra-aortic MMP-9 activity and MMP-9⁺ cells after Ang II infusion. Accordingly, we modified the sections of Results (page 11, paragraph 2 to page 12, paragraph 1) and Methods (page 20, paragraph 2; page 21, paragraph 2) with new Figure 8 and new Figure S7.

2. *The elastin staining is not clear as is. Higher magnification will better depict elastin breaks.*

In accordance with the comment, we added the photos of EVG staining with high magnification and shows new Figure 8e (bottom panel).

3. Did MMP9 levels change in angiotensin II model?

As mentioned above, we examined intra-aortic MMP-9 activity and MMP-9⁺ cells after Ang II infusion. Accordingly, we modified the section of Results (page 12, paragraph 1) with new Figure 8e and new Figure S7b to d).

Minor points

1. *The authors did not change the statement that “Abdominal aortic aneurysm (AAA) results from atherosclerotic changes...”. This sentence is still in the second paragraph of the introduction. As I commented before, it is wrong and misleading to state that all AAA arise from atherosclerotic lesions.*

In accordance with the comment, we modified the description in the revised version (page 3, paragraph 2).

2. *The simplicity of the schematic illustrating their findings in the last supplementary figure needs improvement. It is confusing and unclear as is.*

In accordance with the comment, we improved the last supplementary figure and show new Figure S10.

Reviewer #2 (Remarks to the Author):

My concerns were related to quality of data on the MMPs and relevance of the models to human AAA. The authors have added new data regarding the MMPs, extended and the confirmed similar results in a second animal model and analyzed some human samples. This final analysis confirmed expression of CCR3 and 5 in macrophages in human AAA tissue.

Thank you for your critical and instructive comments.

REVIEWERS' COMMENTS:

Reviewer #1 (Remarks to the Author):

The authors have addressed all my previous concerns.

Note: The reference for the CaCl₂ animal model used is incorrect. It is actually a human study. I would encourage the authors to carefully check whether their bibliographical citations appropriately match the text.

REVIEWERS' COMMENTS:

Reviewer #1 (Remarks to the Author):

The authors have addressed all my previous concerns.

Note: The reference for the CaCl₂ animal model used is incorrect. It is actually a human study. I would encourage the authors to carefully check whether their bibliographical citations appropriately match the text.

Point-by-point reply

Reviewer #1

Note: The reference for the CaCl₂ animal model used is incorrect. It is actually a human study. I would encourage the authors to carefully check whether their bibliographical citations appropriately match the text.

Re: In accordance with the comment, we have corrected reference number (No. 54→ No. 52) in the section of CaCl₂-induced AAA model of Methods of the final version.

Thank you for your critical and instructive comments.